# Iterative Machine Teaching for Black-Box Markov Learners

## Abstract

Machine teaching has traditionally been constrained by the assumption of a fixed learner model, where the learner's progress follows given rules, such as gradient update with fixed learning rates and version space update with a given preference function. In this paper, we consider a generic setting which views the learner as a black box, and the learner's dynamics can be learned during the teaching process. We model the learner's dynamics as a Markov decision process (MDP) with unknown parameters, encompassing a wide range of learner types studied in the machine teaching literature. In such a setting, machine teaching reduces to finding an optimal policy for the underlying MDP. We then introduce an algorithm for teaching such black-box Markov learners, and provide an analysis of the teaching cost under both discounted and non-discounted settings. The Markov learners considered in this work can be naturally linked to epiphany learning as studied in decision psychology. Supported by numerical study results, this paper delivers a novel perspective for machine teaching under the black-box setting, introducing a robust, versatile learner model with a rigorous theoretical foundation.

## 1 Introduction

Machine teaching seeks effective policies for selecting training examples to help a learner learn a target concept. Over the past few decades, the field of machine teaching has been pushed forward and shown great promise in various application domains, including those targeting human learners, such as automated tutoring systems (Rafferty et al., 2016; Sen et al., 2018; Zhu et al., 2018; Hunziker et al., 2019), citizen science and crowdsourcing services (Sullivan et al., 2009; Nugent, 2018), or those targeting machine learning systems, such as model compression (Romero et al., 2014) and understanding the vulnerability of data poisoning attacks (Mei and Zhu, 2015; Zhu, 2018).

As illustrated in figure 1, a machine teaching framework assumes a computational model of the learner—either known or unknown to the teacher—which typically consists of two components: (a) a model for representing the learner's state (e.g., learner's current hypotheses, as in figure 1 (a)), and (b) a model for the learning dynamics (e.g. parameters capturing learner's initial knowledge, learning rate, and learning behavior etc. as in figure 1 (b)). When both models are known to the teacher, the teaching problem boils down to an optimal planning problem as in figure 1 (c). Upon receiving the teaching instructions, the learner makes an update according to its own intrinsic dynamics, and proceeds to the next knowledge state.

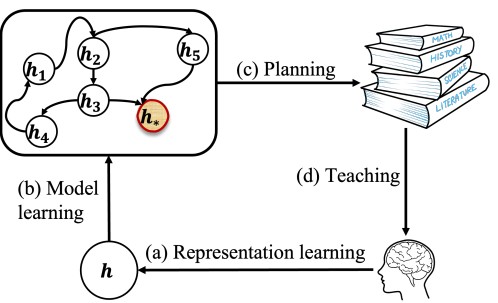

Figure 1: An illustration of the teaching framework. We focus on steps (b), (c) and (d), and assume the feature mapping (i.e. learned through step (a)) is known and given.

Classical theory of machine teaching often focuses on specific realizations of such a framework. Depending on the learner type and how much information of the learner the teacher can access, various teaching models have been proposed. We summarize a few

| model | representation | |
| | known | unknown |
| --- | --- | --- |
| *known* | white-box (Goldman and Kearns, 1995; Zilles et al., 2011; Chen et al., 2018; Mansouri et al., 2019; Lessard et al., 2019; Tabibian et al., 2019; Hunziker et al., 2019) | "black-box learner" (Dasgupta et al., 2019; Liu et al., 2018) |
| *unknown* | black-box MDP learner **(this work)** | – |

Table 1: A summary of different teaching settings and difference between our work and the existing literature

representative works in table 1. Under the "white-box" setting where the teacher has full access to the learner's dynamics and state representation, one may derive strong theoretical guarantees on the complexity of teaching (Goldman and Kearns, 1995; Zilles et al., 2011; Chen et al., 2018; Mansouri et al., 2019; Lessard et al., 2019; Tabibian et al., 2019; Hunziker et al., 2019). When the learner's representation is unknown but the learner's dynamics (e.g. learning algorithm) are given, it has been shown that the teacher can efficiently find a set of teaching examples with strong approximation guarantees in finding the optimal set (Dasgupta et al., 2019) or convergence guarantees (Liu et al., 2018) in teaching the target concept. However, the practical teaching scenario with unknown learner dynamics has been under-explored so far.

To capture the learner's dynamics, we propose to model the learning/teaching problem via a Markov decision process (MDP), where the learner transits among different hypotheses (states) upon receiving teaching instructions (actions). The goal of teaching is to steer the learner towards the goal state (e.g. the concept being taught) via the underlying MDP. As later discussed in section 2, we show that the learner models in table 1 can be viewed as special cases of *Markov learners*. Note that the corresponding studies in the literature are often focused on heuristic learner models or representations. In contrast, the Markov learner entails a versatile framework generic to a broad class of learner models.

Furthermore, most of the existing learner models in algorithmic machine teaching, such as the preference-based version space model (Mansouri et al., 2019; Gao et al., 2017) or the gradient-based model (Liu et al., 2017), assume that the learner follows specific incremental hypothesis update rules which do not capture certain drastic transitions between hypotheses. These learner models naturally align with the concept of non-epiphany learners (Chen and Krajbich, 2017; Dufwenberg et al., 2010), a class of learner studied in decision psychology and neuroscience that was shown not always suitable for modeling human behavior. The restrictions on the hypothesis update rule hinder their applicability to solving practical problems, where the learner model is often a complicated black box (e.g. inferred from historic student data (Corbett and Anderson, 1994; Yudelson et al., 2013; Piech et al., 2015; Settles and Meeder, 2016; Sen et al., 2018; Hunziker et al., 2019)).

**Our contributions.** In this paper, we set forward a generic teaching framework capable of capturing unknown complex learner dynamics in real-world teaching applications. For better understanding of the overall teacher/learner process, we study machine teaching under a generic black-box setting, where the learner's dynamics are modeled by a Markov decision process (MDP) with unknown parameters. We show that many different learner models can be interpreted as Markov learners, and teaching such learners amounts to identifying the optimal policy for the underlying MDP. To provide a theoretical understanding, we derive the teaching cost under the assumption that the learning dynamics can be approximated by a linear function of the learner's state and the teaching instruction. These results are further backed up by a numerical case study demonstrating the effective of the proposed algorithm.

Our contributions are highlighted below.

- We introduce a generic machine teaching model with parametric Markov learners, which can be used in place of many existing learner models for characterizing learner's transition dynamics. This model allows us to estimate the learner's dynamics from data, providing a versatile approach to machine teaching.

- As a side product of our model, we establish a natural connection between Markov learners and the concepts of epiphany and non-epiphany learning in the behavioral science and educational research.

- Under our teaching framework, we provide rigorous analyses on the teaching costs for various teaching scenarios. When the learning dynamics is linear, we show that the teaching costs grows at most polynomially in the optimal teaching cost and feature dimension $d$; when the dynamics is non-linear, we show that teaching is not always feasible, and provide teachability conditions such that the teaching cost becomes linear (ignoring log factors) in the optimal cost.

- Complementing our theoretical results, we conduct conducted a case study on a numerical example to demonstrate the use case of our proposed algorithm, and provide guidelines for setting its hyperparameters.

## 2 Related Work

**Markov learners in machine teaching.** Various learner models studied in the machine teaching literature can be viewed as Markov learners. Among these models, most are investigated under a white-box setting, where both the states $s_t$ (learner's knowledge) and the transition probabilities $\mathbb{P}_\theta$ (learning dynamics) are observable and given to the teacher (Liu et al., 2017; Lessard et al., 2019; Tabibian et al., 2019; Hunziker et al., 2019; Chen et al., 2018; Mansouri et al., 2019). Notably, some recent work consider the black-box setting, where learner's states $s_t$ is unknown but the transition probabilities $\mathbb{P}_\theta$ are given (Dasgupta et al., 2019; Liu et al., 2018; Yudelson et al., 2013; Rafferty et al., 2016). Additionally, popular models in educational research, such as Deep Knowledge Tracing (DKT) (Piech et al., 2015), capture the learner's hypothesis temporal neural networks, typically using Recurrent Neural Networks (RNNs). This setup can conceptually be aligned with the Markov framework by interpreting the decision process as a Partially Observable Markov Decision Process (POMDP), where the states are represented by RNNs. In this model, transition probabilities are derived from empirical data, and the belief states correspond to the cumulative observation history. In section 3, we provide a few more concrete examples, along with their instantiation of the states, actions and transition dynamics under the MDP model. Nevertheless, these machine teaching models all assume the learner's model is known, and are designed in an ad-hoc way. In contrast, our work focuses on proposing a generic framework that can capture these heuristic models and allow learning the learner's model from data.

**Reinforcement learning.** Our work is also closely related to the reinforcement learning literature (Jaksch et al., 2010; Jin et al., 2020; Zhou et al., 2021; Ouyang et al., 2019; Min et al., 2021). In particular, our algorithm design is built upon the least-squares regression algorithm for estimating the parameter of the dynamics function, and the extended value iteration (EVI) (Jaksch et al., 2010) for generating the teaching policy. These two sub-algorithms are commonly used as backbones in algorithm design. In comparison, our work focuses on the non-episodic setting and takes the initialization into consideration, which better fits in the machine teaching problem. Specifically, the learners of interest to this work are always resource-constrained (e.g. by the perceptual capacity of human learning), and such initialization plays a critical role in the final teaching cost as detailed in section 5. Another related line of works is teaching with reinforcement learning policy (Wu et al., 2018; Fan et al., 2018; Florensa et al., 2018; Omidshafiei et al., 2019). However, all of these works focus on improving the training efficiency of neural networks, i.e., whitebox learners. Their major contributions are developing better state representation and reward shaping functions based on different heuristics, which can serve as a complement to our work, i.e., the step (a) in figure 1.

**Epiphany learning.** The concept of Epiphany Learning (EL) has been rigorously studied in behavioral science (Chen and Krajbich, 2017; Dufwenberg et al., 2010). EL denotes a phenomenon where a learning agent (for instance, humans) experiences an abrupt enlightenment or comprehension regarding a specific subject matter. In the context of educational research, EL manifests when students achieve an insightful moment of comprehension or forge a substantial link between concepts, resulting in profound understanding of a topic or problem. Conversely, Non-Epiphany Learning implies scenarios in which such transformative moments do not transpire. Such epiphany/ non-epiphany learners naturally fit into the Markovian framework considered in this paper (see figure 2). We use the MDP learner model as a computational model to capture these learners, and subsequently provide an in-depth analysis of the teaching performance.

| Type of the learner | States | Actions | Transition function |
|---|---|---|---|
| Preference-based version space (Mansouri et al., 2019) | $\boldsymbol{h}_t \in 2^{\mathcal{H}} \times \mathcal{H}$ | $\boldsymbol{x}_t \in \mathcal{X}$ | $\boldsymbol{h}_{t+1} \leftarrow \sigma(\boldsymbol{h}_t, \boldsymbol{x}_t)$ |
| Gradient-based (Liu et al., 2017) | $\boldsymbol{h}_t \in \mathbb{R}^d$ | $\boldsymbol{x}_t \in \mathcal{X}$ | $\boldsymbol{h}_{t+1} \leftarrow \boldsymbol{h}_t - \eta \cdot \nabla_{\boldsymbol{w}} \ell(\boldsymbol{h}_t, \boldsymbol{x}_t)$ |
| Skill-based (Whitehill and Movellan, 2017) | $\boldsymbol{h}_t \in [0,1]^d$ | $\boldsymbol{x}_t \in \mathcal{X}$ | $\boldsymbol{h}_{t+1} \propto \boldsymbol{h}_t \odot q(\boldsymbol{x}_t)^{\alpha} \odot (1 - q(\boldsymbol{x}_t))^{(1-\alpha)}$ |
| Memory-based (Settles and Meeder, 2016; Hunziker et al., 2019) | $\boldsymbol{h}_t \in \mathbb{R}^2$ | $\boldsymbol{x}_t \in \{0,1\}$ | $\boldsymbol{h}_{t+1} \leftarrow \boldsymbol{h}_t \cdot \mathrm{HL}(\boldsymbol{x}_t)$ |

Table 2: Examples of existing sequential learner models that have the Markov property. Detailed discussion over these models is provided in section 3.1.

## 3 Problem Formulation

We deal with the black-box setting, where the teacher initially has no knowledge of the learning dynamics (e.g., the parameters) of the learner, i.e., how the learner updates its knowledge state upon receiving the teacher instruction. We assume that the teacher can observe the learner's state directly, and also knows the cost function[1]. The goal of the teacher is to help the learner reach some target knowledge state with minimal cost. To assist the learner, the teacher will not only provide informative teaching instructions to the learner but also needs to learn about the learner's dynamics.

**Notations.** Before we proceed, we first introduce some notation. We use $\mathcal{H}$ to represent the set of all possible knowledge states of learners, $\boldsymbol{h}_0$ denotes the initial knowledge state, and $\boldsymbol{h}_t$ is the learner's knowledge state at iteration $t$. At each iteration $t$, the teacher can choose one teaching instruction $\boldsymbol{x}_t$ from the teaching set $\mathcal{X}$. The learner's knowledge state will be updated upon receiving the teaching instruction. The teacher's goal is to help the learner transit to the target knowledge state $\boldsymbol{h}^{\star}$ with minimal cost. Throughout the entire paper, we use $C_{\star}$ to denote the tightest upper bound on the expected teaching cost of the optimal teaching policy by starting from any initial state.

### 3.1 Parametric Markov Learners

We model the learner as a Markov learner, which is able to cover a broad class of learners considered in the literature (Gao et al., 2017; Whitehill and Movellan, 2017; Liu et al., 2017; Hunziker et al., 2019; Mansouri et al., 2019). Specifically, for any given learner, it starts from some initial knowledge state $\boldsymbol{h}_0$, which represents its current knowledge state. For each iteration $t$, when the learner receives the teaching instruction $\boldsymbol{x}_t$, it updates its knowledge based on its transition probability,

$$\boldsymbol{h}_{t+1} \sim \mathbb{P}_{\boldsymbol{\theta}^{\ell}}[\boldsymbol{h}_{t+1} | \boldsymbol{h}_t, \boldsymbol{x}_t], \tag{1}$$

where $\boldsymbol{\theta}^{\ell}$ refers to the parameters that define the transition probability or learning dynamics of the learner. Different learners may have different $\boldsymbol{\theta}^{\ell}$. The transition probability induces a preference over the next knowledge states for the learner, which captures the learning dynamics of the learner. Intuitively, for fast learners, it will assign a higher transition probability to states that are close to the target state $\boldsymbol{h}^{\star}$ upon receiving the teaching instructions. In contrast, sometimes, a learner may not be able to understand advanced teaching instructions before it reaches certain knowledge state. To model such scenarios, the learner may assign a very high probability to remain at the current knowledge state when receiving obscure teaching instructions (i.e., no learning progress after receiving the teaching instruction).

In the following, we will illustrate through a set of examples how the parametric Markov model described in equation 1 captures the learner's dynamics characterized by several existing sequential learner models, as summarized in table 2.

**Example 1 (Preference-based model for version space learner)** *For preference-based learners (Chen et al., 2018; Mansouri et al., 2019), a state $\boldsymbol{h}_t := (\mathcal{H}_t, h_t)$ is represented as a combination of the learner's current version space, denoted by $\mathcal{H}_t$, and their current hypothesis, denoted by $h_t$. This representation captures both the set of all hypotheses that are consistent with the observed data (the version space) and the learner's*

---

[1]In practice, the teacher can probe the learner's knowledge state by quizzing the learner. The cost could be the price of the teaching instruction.

*specific hypothesis at any given time. An action $\boldsymbol{x}_t$ corresponds to the provision of a teaching example, which influences the learner's hypothesis. The transition from one state to the next is governed by the preference function $\sigma$, determining how the learner navigates through $\mathcal{H}_t$ in response to teaching actions:*

$$\boldsymbol{h}_{t+1} \leftarrow \sigma(\boldsymbol{h}_t, \boldsymbol{x}_t)$$

Mansouri et al. (2019) show that by instantiating the preference-based learner with different preference functions $\sigma$, it reduces to several classic learner models in algorithmic machine teaching: When $\sigma(\boldsymbol{h}_t, \boldsymbol{x}_t) = c$ for some constant $c$, it corresponds to the classic "worst-case" version space model (Goldman and Kearns, 1995); when $\sigma(\boldsymbol{h}_t, \boldsymbol{x}_t) = g(\cdot)$ for some function $g$ that does not depend on the learner's current state (i.e. current hypothesis $h_t$ and the version space $\mathcal{H}_t$, it corresponds to the (global) preference-based learner's model (Zilles et al., 2011; Gao et al., 2017).

**Example 2 (Gradient-based learner)**

$$\boldsymbol{h}_{t+1} \leftarrow \boldsymbol{h}_t - \eta \cdot \nabla_{\boldsymbol{w}} \ell(\boldsymbol{h}_t, \boldsymbol{x}_t)$$

*where $\eta$ denotes the learning rate, $\boldsymbol{h}_t \in \mathbb{R}^d$ denotes the learner's state at $t$, and $\ell$ denotes the loss function.*

Liu et al. (2017; 2018) study both the white-box setting and the black-box setting. For the black-box setting, they assume that the learner's state $\boldsymbol{h}_t$ is unknown but the transition function, defined through the learning rate $\eta$, is known. When $\eta$ is unknown, the teacher needs to spend an extra budget to estimate it, which does not affect the teaching complexity overall.

**Example 3 (Skill-based learner)** *Skill-based learners (Bower, 1961; Corbett and Anderson, 1994; White-hill and Movellan, 2017) conceptualize learning as the acquisition of discrete, independent skills. In the simplest form of such models, the state space corresponds to $d$ independent skills; each skill is binary, indicating whether it is "learned" (1) or "not learned" (0). At time step $t$, when an exercise $\mathbf{x}_t$ is presented, the skill associated with $\mathbf{x}_t$ can jump from 0 to 1 state with some probability. This probabilistic transition is captured by*

$$\boldsymbol{h}_{t+1} \propto \boldsymbol{h}_t \odot q(\boldsymbol{x}_t)^{\alpha} \odot (1 - q(\boldsymbol{x}_t))^{(1-\alpha)}$$

*Here, $q(\boldsymbol{x}_t)$ represents the probability of learning the skill associated with exercise $\boldsymbol{x}_t$, and $\alpha$ is a binary variable indicating the presence (1) or absence (0) of the skill prior to $\boldsymbol{x}_t$.*

Skill-based learners represent a fundamental learner's model in intelligent tutoring systems (ITS), which is integral to the knowledge tracing frameworks (Corbett and Anderson, 1994). More advanced models such as Deep Knowledge Tracing (Piech et al., 2015) extend beyond binary skill states, employing continuous and correlated variables to capture more intricate representation of learners' skill sets, thereby enhancing the model's capacity to navigate and support the complex landscape of learning trajectories.

**Example 4 (Memory-based learner)** *Classical computational models of human memory, such as the Half-Life Regression (HLR) model (Settles and Meeder, 2016), have been used in machine teaching to model the long term learning behavior of a human subject. The HLR model posits an exponential decay of memory over time, where the probability of correctly recalling an item is influenced by the time elapsed since last reviewed, and the memory strength, quantified as the half-life ($HL_\theta(\cdot)$). A state $\boldsymbol{h}_t$ corresponds to a retention level and a forgetting rate, and a transition is specified by the half-life dynamics HL:*

$$\boldsymbol{h}_{t+1} \leftarrow \boldsymbol{h}_t \cdot HL_\theta(\boldsymbol{x}_t).$$

A concrete HLR model studied by Settles and Meeder (2016) calculates the half-life based on a learner's interactions with the teaching example, using the feature representation $\boldsymbol{x}_t$ and a parameter vector $\theta$, yielding the estimated half-life as $HL_\theta(\boldsymbol{x}_t) = 2^{\theta \cdot \boldsymbol{x}_t}$. This model extends beyond traditional methods like Leitner (Leitner and Totter, 1972) and Pimsleur (Pimsleur, 1967) systems by empirically fitting $\theta$ to actual learning data, accommodating a wider array of features to more accurately mirror a learner's memory dynamics.

### 3.2 The Teacher's Objective

The teacher's goal is to help the learner learn as fast as possible, i.e., minimizing the cost of steering the learner to reach the target knowledge state $\boldsymbol{h}^\star$. In order to teach, there are two tasks that the teacher needs to solve, namely estimating $\boldsymbol{\theta}^\ell$ and generating the teaching instruction. The entire problem can be formulated as follows, where $c(\cdot, \cdot)$ is the cost function.

$$\min_{\boldsymbol{x}_{1:T} \in \mathcal{X}^T, T \in \mathbb{Z}_+} \quad \mathbb{E}\left[\sum_{t=1}^{T} c(\boldsymbol{h}_t, \boldsymbol{x}_t)\right] \quad s.t. \quad \boldsymbol{h}_{T+1} = \boldsymbol{h}^\star \text{ and } \boldsymbol{h}_{t+1} \sim \mathbb{P}_{\boldsymbol{\theta}^\ell}(\boldsymbol{h}|\boldsymbol{h}_t, \boldsymbol{x}_t). \tag{2}$$

If the teacher knows the true parameters, then the above problem becomes a (stochastic) planning problem. In this work, we assume that the teacher only knows the parametric form of the learner's transition function, and it doesn't know the true parameters of the learner. This makes our problem formulation more general, but also introduces an extra challenge in solving the above problem.

## 4 Preliminaries and Background

Teaching Markov learners can be captured by an MDP $M := \{\mathcal{H}, \mathcal{X}, \mathbb{P}, c, \boldsymbol{h}_0, \boldsymbol{h}^\star, \gamma\}$, where $c : \mathcal{H} \times \mathcal{X} \to \mathbb{R}_+$ is the cost function and $\boldsymbol{h}^\star$ is the target knowledge state. For any $(\boldsymbol{h}, \boldsymbol{x}, \boldsymbol{h}') \in \mathcal{H} \times \mathcal{X} \times \mathcal{H}$, $\mathbb{P}_{\boldsymbol{\theta}^\ell}(\boldsymbol{h}'|\boldsymbol{h}, \boldsymbol{x})$ denotes the probability of transiting to knowledge state $\boldsymbol{h}'$ given the teaching instruction $\boldsymbol{x}$ under $\boldsymbol{h}$. To be noted, when the learner reaches the target knowledge state, the cost will be zero for all the teaching instructions, i.e., $c(\boldsymbol{h}^\star, \boldsymbol{x}) = 0, \forall \boldsymbol{x} \in \mathcal{X}$, and $\mathbb{P}(\boldsymbol{h}^\star|\boldsymbol{h}^\star, \boldsymbol{x}) = 1$, which means the target knowledge state is an absorbing state. $\gamma \in (0, 1]$ is the cost discounting factor. In the teaching context, $1 - \gamma$ is the probability of the learner transiting to the target knowledge state from any other state, i.e., the probability of epiphany learning (Dufwenberg et al., 2010; Chen and Krajbich, 2017).

**Definition 1 (Proper Policy)** *A stationary policy $\pi$ is proper if, given any initial state, the probability of reaching the goal state $g$ within a finite number of steps, when following $\pi$, is strictly positive.*

In the remaining of this section, we introduce the key assumptions that the subsequent sections rely on. Let us denote by $\Pi_{\text{proper}}(M)$ the set of stationary polices of the underlying MDP $M$ such that for any policy $\pi \in \Pi_{\text{proper}}(M)$, the expected time that it takes to reach the target knowledge state $\boldsymbol{h}^\star$ from any initial knowledge state $\boldsymbol{h}$ is finite. In the teaching context, the existence of proper polices for a learner means that there is a way to teach him/her the target knowledge state $\boldsymbol{h}^\star$. In our analysis, we will assume that the Markov learner is linear and teachable under some known and given feature mapping $\boldsymbol{\phi} : \mathcal{H} \times \mathcal{X} \times \mathcal{H} \to \mathbb{R}^d$. We summarize the essential idea in the following assumption. Similar assumption has also been studied in Zhou et al. (2021); Min et al. (2021).

**Assumption 1 (Teachable Linear Markov Learners)** $M := \{\mathcal{H}, \mathcal{X}, \mathbb{P}_{\boldsymbol{\theta}^\ell}, c, \boldsymbol{h}_0, \boldsymbol{h}^\star, \gamma\}$ *is a teachable linear Markov learner, if it satisfies*

- ***Linearity:*** *Given a known feature mapping $\boldsymbol{\phi}$, there exists an unknown parameter $\boldsymbol{\theta}^\ell \in \mathbb{R}^d$ ($\|\boldsymbol{\theta}^\ell\|_2^2 \leq d$) such that $\mathbb{P}_{\boldsymbol{\theta}^\ell}(\boldsymbol{h}'|\boldsymbol{h}, \boldsymbol{x}) = \langle \boldsymbol{\phi}(\boldsymbol{h}'|\boldsymbol{h}, \boldsymbol{x}), \boldsymbol{\theta}^\ell \rangle, \forall (\boldsymbol{h}, \boldsymbol{x}, \boldsymbol{h}') \in \mathcal{H} \times \mathcal{X} \times \mathcal{H}$.*

- ***Teachable:*** *There exists at least one proper policy, i.e., $\Pi_{proper}(M) \neq \emptyset$.*

*Furthermore, for any bounded value function $V : \mathcal{H} \to [0, C]$ with $C_\star \leq C$, $\|\boldsymbol{\phi}_V(\boldsymbol{h}, \boldsymbol{x})\|_2 \leq \sqrt{d}C$ holds for any $(\boldsymbol{h}, \boldsymbol{x}) \in \mathcal{H} \times \mathcal{X}$, where $\boldsymbol{\phi}_V(\boldsymbol{h}, \boldsymbol{x}) = \sum_{\boldsymbol{h}'} \boldsymbol{\phi}(\boldsymbol{h}'|\boldsymbol{h}, \boldsymbol{x})V(\boldsymbol{h}')$.*

For any value function $V : \mathcal{H} \to \mathbb{R}_+$, we define $\mathbb{P}V(\boldsymbol{h}, \boldsymbol{x}) = \sum_{\boldsymbol{h}'} \mathbb{P}(\boldsymbol{h}'|\boldsymbol{h}, \boldsymbol{x})V(\boldsymbol{h}')$ for any $(\boldsymbol{h}, \boldsymbol{x}) \in \mathcal{H} \times \mathcal{X}$. Under the linear MDP assumption, we further have $\mathbb{P}_{\boldsymbol{\theta}^\ell}V(\boldsymbol{h}, \boldsymbol{x}) = \langle \boldsymbol{\phi}_V(\boldsymbol{h}, \boldsymbol{x}), \boldsymbol{\theta}^\ell \rangle$. For convenience of notation, we further define the cost-to-go function for policy $\pi$ under $M_{\boldsymbol{\theta}^\ell}$ as

$$V^\pi(\boldsymbol{h}|\boldsymbol{\theta}^\ell) := \lim_{T \to +\infty} \mathbb{E}\left[\sum_{t=0}^{T} c(\boldsymbol{h}_t, \boldsymbol{x}_t) \middle| \boldsymbol{h}_0 = \boldsymbol{h}\right], \text{ where } \boldsymbol{h}_{t+1} \sim \mathbb{P}_{\boldsymbol{\theta}^\ell}(\boldsymbol{h}|\boldsymbol{h}_t, \boldsymbol{x}_t) \text{ and } \boldsymbol{x}_t = \pi(\boldsymbol{h}_t).$$

Consequently, the Q-value function of policy $\pi$ under $M_{\boldsymbol{\theta}^\ell}$ can be written as

$$Q^\pi(\boldsymbol{h}, \boldsymbol{x}|\boldsymbol{\theta}^\ell) := c(\boldsymbol{h}, \boldsymbol{x}) + \mathbb{P}_{\boldsymbol{\theta}^\ell}V^\pi(\boldsymbol{h}, \boldsymbol{x}|\boldsymbol{\theta}^\ell). \tag{3}$$

Subsequently, when there is no ambiguity, we use $V(\boldsymbol{h})$ and $Q(\boldsymbol{h}, \boldsymbol{x})$ to simplify notation. Next, we introduce another assumption tailored to the teaching setting.

**Assumption 2 ($\delta_0$-Closeness)** *The true parameter $\boldsymbol{\theta}^\ell$ is $\delta_0$-close to the teacher's initial estimation $\boldsymbol{\theta}_0$, i.e., $\|\boldsymbol{\theta}^\ell - \boldsymbol{\theta}_0\|_2 \leq \delta_0 \sqrt{d}$ with $0 \leq \delta_0 \leq 1$.*

The above assumption is natural in the teaching setting. Without such an assumption, the teacher may need to interact with the learner for a large number of rounds before it can teach in an effective way, which is impractical for teaching resource-constrained learners, such as humans. In practice, to fulfil Assumption 2, we can first fit a transition function on the offline teacher-learner interaction data to serve as the initialization. For simplicity, we denote the associated MDP of a learner with parameter $\boldsymbol{\theta}^\ell$ as $M_{\boldsymbol{\theta}^\ell}$ and the teacher's initial estimation on the parameter as $\boldsymbol{\theta}_0$.

Lastly, we define two categories of Markov learners depending on their behaviors during learning, which can be modelled by undiscounted MDP and discounted MDP, respectively. We call a Markov learners an *epiphany learner* if $\gamma < 1$ for its associated MDP. When the learner's associated MDP has $\gamma = 1$, we call it a *non-epiphany learner*.

Epiphany learning (Dufwenberg et al., 2010; Chen and Krajbich, 2017) is an observed phenomenon in human learners, which refers to sudden insights or realizations of human learners that lead to a rapid increase in understanding or problem-solving ability. Such learners may not show gradual improvement but instead have significant leaps in learning after periods of stagnation or slow progress. In the context of machine teaching (see figure 2), the 'epiphany' or sudden leap in understanding can be viewed as a significant reward. More specifically, such epiphany is specifically modeled as direct transitions to the *goal state*, highlighting its significant impact on the learning process. The discount factor $1-\gamma$ could model the decreasing likelihood or value of such epiphanies over time, or the increasing value of immediate, incremental learning compared to waiting for less predictable, more significant breakthroughs. In other words, $1-\gamma$ can be intuitively understood as the lower bound of the probability of epiphany learning (i.e., transit to the goal knowledge state) at all the knowledge states. The skill-based learners in table 2 can naturally be considered as epiphany learners, while the others (e.g., preference-based learners,

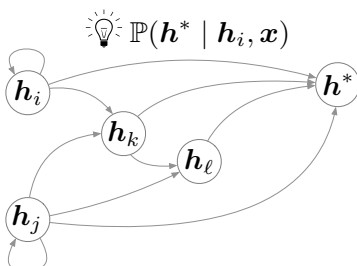

Figure 2: Modeling epiphany learning as a discounted MDP. The 'epiphany' or sudden leap is illustrated by the dashed arrows with a light bulb. The solid arrows represent normal transitions between states. The probability of epiphany at state $\boldsymbol{h}_i$ can be interpreted as the probability of reaching the goal state $\mathbb{P}(\boldsymbol{h}^* \mid \boldsymbol{h}_i, \boldsymbol{x})$.

gradient learners and memory-based learners) are more suitable to be modeled as non-epiphany learners.

## 5 Teaching Black-box Markov Learners: Algorithm and Analysis

In this section, we present an algorithm for teaching black-box Markov learners (including epiphany learners and non-epiphany learners), which takes the initialization into account. We then conduct a rigorous analysis for upper bounding the teaching cost under different teaching scenarios, including 1) the Markov learner is linear and teachable; 2) the Markov learner is nonlinear and teachable.

### 5.1 Black-box Teaching for Linear Markov Learners: Algorithm

We first consider the case where the Markov learner is linear and teachable (see Assumption 1). We first present an algorithm for solving the teaching problem, which takes the initialization into consideration. The entire algorithm is built upon solving a regularized least-squares regression (for computing $\hat{\boldsymbol{\theta}}$), and extended value iteration (for generating the teaching policy). These two sub-algorithms are often used as backbones designing RL algorithms (Jaksch et al., 2010; Jin et al., 2020; Zhou et al., 2021; Ouyang et al., 2019; Min et al., 2021). In contrast to these works, our algorithm 1) takes the initialization $\boldsymbol{\theta}_0$ into account, which

---

**Algorithm 1** Black-box Teaching Algorithm for Non-Epiphany and Epiphany Learners.

---

**Require:** Initial estimation $\hat{\boldsymbol{\theta}}_0 = \boldsymbol{\theta}_0$, iteration $t = 0$, EVI index $t_0 = 0$, $k = 0$, $\boldsymbol{\Sigma}_0 = \lambda\boldsymbol{I}$, $\boldsymbol{\mu}_0 = \lambda\boldsymbol{\theta}_0$ and
    discount factor $\gamma$ (for epiphany learners).
1: $Q_0 \leftarrow EVI(\{\boldsymbol{\theta} \in \mathbb{R}^d \mid \|\boldsymbol{\theta} - \hat{\boldsymbol{\theta}}_0\|_2 \leq \delta_0\}, \frac{1}{\lambda}, \frac{1}{\lambda})$
2: **while** $\boldsymbol{h}_t \neq \boldsymbol{h}^\star$ **do**
3:     Provide $\boldsymbol{x}_t = \arg\min_{\boldsymbol{x} \in \mathcal{X}} Q_k(\boldsymbol{h}_t, \boldsymbol{x})$
4:     Receive $c_t = c(\boldsymbol{h}_t, \boldsymbol{x}_t)$; $\boldsymbol{h}_{t+1} \sim \mathbb{P}_{\boldsymbol{\theta}^\ell}(\cdot|\boldsymbol{h}_t, \boldsymbol{x}_t)$.
5:     $\boldsymbol{\Sigma}_t \leftarrow \boldsymbol{\Sigma}_{t-1} + \boldsymbol{\phi}_{V_k}(\boldsymbol{h}_t, \boldsymbol{x}_t)\boldsymbol{\phi}_{V_k}(\boldsymbol{h}_t, \boldsymbol{x}_t)^\top$
6:     $\boldsymbol{\mu}_t \leftarrow \boldsymbol{\mu}_{t-1} + \boldsymbol{\phi}_{V_k}(\boldsymbol{h}_t, \boldsymbol{x}_t)V_k(\boldsymbol{h}_{t+1})$
7:     **if** $\det(\boldsymbol{\Sigma}_t) \geq 2\det(\boldsymbol{\Sigma}_{t_k})$ or $t \geq 2t_k + \lambda$ **then**
8:         $k \leftarrow k + 1$
9:         $t_k \leftarrow t$
10:        $Q_k = EVI(\mathcal{C}_t, \frac{1}{\lambda t}, 1 - \frac{1}{\lambda t})$
11:        $Q_k = EVI(\mathcal{C}_t, \frac{1}{\lambda t}, \gamma)$
12:     **end if**
13:     $t \leftarrow t + 1$
14: **end while**

---

is crucial to teaching effectively; 2) and applies to both epiphany and non-epiphany learners. Intuitively, Algorithm 1 can be divided into two parts as described below.

**Parameter learning.** For parameter learning, once the updating criteria is satisfied, the teacher will update its estimation of the learner's parameter based on the interactions so far. Updating the estimation reduces to solving the following initialization-regularized least-squares problem:

$$\hat{\boldsymbol{\theta}}_m \leftarrow \arg\min_{\boldsymbol{\theta} \in \mathbb{R}^d} \sum_{t=0}^{m-1} [\langle \boldsymbol{\phi}_{V_{k(t)}}(\boldsymbol{h}_t, \boldsymbol{x}_t), \boldsymbol{\theta} \rangle - V_{k(t)}(\boldsymbol{h}_{t+1})]^2 + \lambda\|\boldsymbol{\theta} - \boldsymbol{\theta}_0\|_2^2, \tag{4}$$

where $k(t)$ is the index of the value function at iteration $t$, e.g., for $t_{j-1} \leq t \leq t_j - 1$, the index is $k(t) = j - 1$, and $V_j(h)$ is the $j_{th}$ value function returned by the extended value iteration (EVI) algorithm (Jaksch et al., 2010). The above problem has a closed-form solution $\hat{\boldsymbol{\theta}}_m = \boldsymbol{\Sigma}_m^{-1}\boldsymbol{\mu}_m$, where (see also Lines 4&5 in Algorithm 1),

$$\boldsymbol{\Sigma}_m = \lambda\boldsymbol{I} + \sum_{t=0}^{m-1} \boldsymbol{\phi}_{V_{k(t)}}(\boldsymbol{h}_t, \boldsymbol{x}_t)\boldsymbol{\phi}_{V_{k(t)}}(\boldsymbol{h}_t, \boldsymbol{x}_t)^\top, \quad \boldsymbol{\mu}_m = \lambda\boldsymbol{\theta}_0 + \sum_{t=0}^{m-1} \boldsymbol{\phi}_{V_{k(t)}}(\boldsymbol{h}_t, \boldsymbol{x}_t)V_{k(t)}(\boldsymbol{h}_{t+1}).$$

The value of $\lambda$ indicates our confidence on the optimality of the initialization $\boldsymbol{\theta}_0$. When the initialization is very likely close to the true parameter $\boldsymbol{\theta}^\ell$, we should set a large $\lambda$, otherwise we should set a small $\lambda$. In addition, $\lambda$ also affects the updating frequency of the parameter, which is triggered by two criteria, namely 1) the log-determinant of $\boldsymbol{\Sigma}_t$; and 2) the number of iterations (see Line 7 in Algorithm 1). When $\lambda$ is larger, the parameter will be updated less frequently, since we trust our current estimate more. To be noted, in our analysis, we always assume $\lambda \geq 1$[2].

**Teaching.** During the teaching phase, the teacher's policy is induced by the Q-value function returned by EVI (see Algorithm 2). After the teacher's teaching instruction, the teacher will receive a cost incurred by the teaching instruction, and also observe the learner's latest knowledge state,

$$\boldsymbol{x}_t = \arg\min_{\boldsymbol{x} \in \mathcal{X}} Q_{k(t)}(\boldsymbol{h}_t, \boldsymbol{x}_t), \text{ where } c_t = c(\boldsymbol{h}_t, \boldsymbol{x}_t), \ \boldsymbol{h}_{t+1} \sim \mathbb{P}_{\boldsymbol{\theta}^\ell}(\boldsymbol{h}|\boldsymbol{h}_t, \boldsymbol{x}_t). \tag{5}$$

In detail, the EVI algorithm takes the confidence set $\mathcal{C}_t$ (see Lemma 1 for $t \geq 1$), the error tolerance of the value iteration $\xi$ and the cost discounting factor $\nu$ as input. The confidence set $\mathcal{C}_t$ is an ellipsoid centered at the current estimation $\hat{\boldsymbol{\theta}}_t$. With high probability, the true parameter $\boldsymbol{\theta}^\ell$ lies in the intersection of $\mathcal{B}$ and $\mathcal{C}_t$, where $\mathcal{B}$ is defined as

$$\mathcal{B} := \{\boldsymbol{\theta} \in \mathbb{R}^d \mid \langle \boldsymbol{\phi}(\cdot|\boldsymbol{h}, \boldsymbol{x}), \boldsymbol{\theta} \rangle \in \Delta^d, \ \forall(\boldsymbol{h}, \boldsymbol{x}) \in \mathcal{H} \times \mathcal{X}\}.$$

---

[2]This is because we found that $\lambda \propto 1/\delta_0^2$ is a good choice in practice (see section 6), and $\delta_0 \leq 1$ by Assumption 2.

---

**Algorithm 2** Extended Value Iteration: $\text{EVI}(\mathcal{C}, \xi, \nu)$

---

**Require:** Confidence set $\mathcal{C}$, error tolerance of valute iteration $\xi$, iteration $i = 0$, cost discount factor $\nu$.
1: $Q^{(0)}(\cdot, \cdot) = 0$
2: $Q(\cdot, \cdot) = 0$
3: $V^{(0)}(\cdot) = 0$
4: $V^{(-1)}(\cdot) = +\infty$
5: **if** $\mathcal{C} \cap \mathcal{B} \neq \emptyset$ **then**
6:     **while** $\|V^{(i)} - V^{(i-1)}\|_\infty \geq \epsilon$ **do**
7:         $Q^{(i+1)}(\cdot, \cdot) \leftarrow c(\cdot, \cdot) + \nu \cdot \min_{\boldsymbol{\theta} \in \mathcal{C} \cap \mathcal{B}} \langle \boldsymbol{\theta}, \boldsymbol{\phi}_{V^{(i)}}(\cdot, \cdot) \rangle$
8:         $V^{(i+1)}(\cdot) \leftarrow \min_{\boldsymbol{x} \in \mathcal{X}} Q^{(i+1)}(\cdot, \boldsymbol{x})$
9:         $i \leftarrow i + 1$
10:    **end while**
11:     $Q(\cdot, \cdot) \leftarrow Q^{(i+1)}(\cdot, \cdot).$
12: **end if**
13: return $Q(\cdot, \cdot)$

---

The error tolerance parameter is chosen to be $\xi = 1/(\lambda t)$. Intuitively, the error tolerance will be smaller when we 1) collect more data (i.e., $t$ becomes large); and 2) start from a better initialization (i.e., $\delta_0$ is smaller). For the cost discount factor $\nu$, we set it to be $1 - 1/(\lambda t)$, when the underlying MDP of the Markov learner is undiscounted (i.e., *non-epiphany learners*). By doing so, the cost discount factor $\nu$ will become closer to 1 as the teaching continues, which helps us avoid a teaching cost that is linear in $T$ (i.e, the total number of teaching instructions) and also ensures the convergence of EVI. When the learner's underlying MDP is discounted (i.e., *epiphany learners*), we will set $\nu = \gamma$ to be a constant. Intuitively, the cost discount factor $\nu$ captures the probability of epiphany learning.

Overall, the EVI algorithm adapts the standard value iteration algorithm to incorporate the optimism-in-the-face-of-uncertainty (OFU) principle (see Line 7 in Algorithm 2) proposed by Abbasi-Yadkori et al. (2011), which has been demonstrated to be effective in online learning settings.

## 5.2 Theoretical Analysis for the Linear Case

In this section, we analyze the cost upper bounds of using Algorithm 1 for teaching both non-epiphany and epiphany learners. The core of the algorithm is to build the confidence set that contains the true parameter $\boldsymbol{\theta}^\ell$, which balances exploration (parameter learning) and exploitation (teaching). In general, the smaller the confidence set that we can construct, the lower the cost. In the following, we present Lemma 1, which provides a confidence set containing $\boldsymbol{\theta}^\ell$ with high probability.

**Lemma 1** *Under Assumptions 1 and 2, for any $t \geq 1$, with probability at least $1 - \delta$, we have that the true parameter $\boldsymbol{\theta}^\ell$ lies in*

$$\mathcal{C}_t = \left\{ \boldsymbol{\theta} \in \mathbb{R}^d \,\middle|\, \|\hat{\boldsymbol{\theta}}_t - \boldsymbol{\theta}\|_{\boldsymbol{\Sigma}_t} \leq C\sqrt{d \log\left((4(t^2 + t^3 C^2/\lambda))/\delta\right)} + \sqrt{\lambda}\delta_0 \right\}. \tag{6}$$

The confidence set $\mathcal{C}_t$ is centered at the current estimation $\hat{\boldsymbol{\theta}}_t$. Its radius is computed based on the iteration $t$, feature dimension $d$, regularization parameter $\lambda$, the upper bound of the optimal cost $C$, and the upper bound on the distance between $\boldsymbol{\theta}_0$ and $\boldsymbol{\theta}^\ell$, i.e., $\delta_0$. As expected, the confidence set will become smaller as $\delta_0$ decreases, indicating that a good initialization is desired. Now Theorem 1 provides an upper bound on the cost of teaching non-epiphany learners using Algorithm 1.

**Theorem 1** *Under Assumptions 1 and 2, if the confidence set $\mathcal{C}_t$ is constructed according to Lemma 1 with $C = \mathcal{O}(C_\star)$, $\lambda = 1/\delta_0^2$, and the cost function is bounded from below by $c_{\min}$ for all non-goal knowledge states $(\mathcal{H} \setminus \{\boldsymbol{h}^\star\})$ and teaching instruction $(\mathcal{X})$ pairs, then with probability at least $1 - 2\delta$, the teaching cost of Algorithm 1 for non-epiphany learners (i.e., $\gamma = 1$) is upper bounded by*

$$\mathcal{O}\left( \left( 1 + d\sqrt{\log\left(1 + \frac{C_\star d \delta_0}{\delta c_{\min}}\right)} \right) \cdot \log^{1.5}\left(\frac{C_\star d}{c_{\min}\delta}\right) \cdot \frac{C_\star^2 d}{c_{\min}} \right). \tag{7}$$

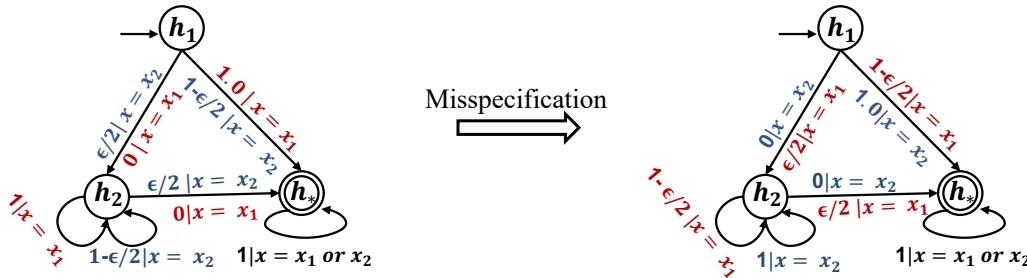

Figure 3: An illustration of the failure case under the misspecified setting. The MDP consists of 2 teaching actions $\mathcal{X} = \{\boldsymbol{x}_1, \boldsymbol{x}_2\}$ and 3 states $\mathcal{H} = \{\boldsymbol{h}_0, \boldsymbol{h}_2, \boldsymbol{h}^\star\}$, and the misspecification level is $\epsilon$. For the teaching policy induced by the misspecified MDP (right), the learner can get stuck at the state $\boldsymbol{h}_2$ with probability $\epsilon/2$.

The cost upper bound in Theorem 1 has a polynomial dependency on the expected cost of the optimal policy, $C_\star$. It's worth noting that when $\delta_0 \to 0$, the purple term inside the parentheses of Equation 7 will vanish leaving only the constant term 1. The constant term 1 is due to the stochasticity in the transition of the learner's knowledge states, which is independent of the teaching algorithm used.

Next, we consider the case where the learner is an *epiphany learner*. Intuitively, epiphany learning can be interpreted as adding a shortcut from the current knowledge state to the target knowledge state in the underlying MDP, which is equivalent to the discounted MDP case. The following theorem provides an upper bound on the cost of teaching epiphany learners.

**Theorem 2** *Under Assumptions 1 and 2, if the confidence set $\mathcal{C}_t$ is constructed according to Lemma 1 with $C = \mathcal{O}(C_\star)$, $\lambda = 1/\delta_0^2$, then with probability at least $1 - 3\delta$, the total cost incurred by running Algorithm 1 for epiphany learners with $\gamma < 1$, is upper bounded by*

$$\mathcal{O}\left(C_\star \cdot \left(1 + d\sqrt{\log\left(1 + \frac{C_\star^2 \delta_0^2 \log\delta}{\log\gamma}\right)}\right) \cdot \sqrt{\frac{\log\delta}{\log\gamma} \log\left(\frac{C_\star \log\delta}{\delta \log\gamma}\right)}\right). \tag{8}$$

Compared with Theorem 1 for non-epiphany learners, the upper bound of the teaching cost for epiphany learners is linear (ignoring the log factors) in the expected teaching cost of the optimal policy $C_\star$ and the feature dimension $d$. Moreover, the dependency on $d$ will vanish when $\delta_0 \to 0$ as well. In addition, Theorem 2 does not require the cost function to be bounded from below.

### 5.3 Theoretical Analysis for the Non-linear Case

In the previous section, we presented the theoretical analysis for both non-epiphany and epiphany learners when their learning dynamics are linear. One natural follow-up question is: what would happen if the learner's dynamics is non-linear, i.e., the linear model is misspecified? To study this problem, we consider the case where teaching the learner can be *approximately* modelled as a linear MDP. This idea is captured in the following assumption.

**Definition 2 ($\epsilon$-Approximate Teachable Markov Learners)** *For any $\epsilon \in (0, 1]$, a MDP $M = (\mathcal{H}, \mathcal{X}, \mathbb{P}, c, \boldsymbol{h}_0, \boldsymbol{h}^\star, \gamma)$ is an $\epsilon$-approximate teachable MDP with a feature map $\boldsymbol{\phi}$, if there exists a unknown teachable linear MDP $M_{\boldsymbol{\theta}^\star}$ such that for any $(\boldsymbol{h}, \boldsymbol{x}) \in \mathcal{H} \times \mathcal{X}$, we have $\|\mathbb{P}(\cdot|\boldsymbol{h}, \boldsymbol{x}) - \langle\boldsymbol{\phi}(\cdot|\boldsymbol{h}, \boldsymbol{x}), \boldsymbol{\theta}^\star\rangle\|_{TV} \leq \epsilon$, ,where TV denotes the total variation distance.*

By definition, the learner is an $\epsilon$-approximate teachable Markov learner if the learning dynamics function of the learner is close to a linear transition function under the given feature mapping $\boldsymbol{\phi}$. We measure the closeness between the dynamics functions by the total variation distance.

In general, the algorithm designed for the linear case will fail when the transition function is non-linear. Specifically, for non-epiphany learners, the teaching cost can be unbounded even for a small model misspecification level $\epsilon$. To illustrate this, we present an informal example (see Figure 3), where the teaching policy

induced by the closest linear MDP to the learner's MDP will incur an infinite teaching cost. The intuition behind such counterexamples is that the teaching policy induced by the misspecified MDP will get trapped in a circle of the true MDP. Fortunately, for epiphany learners, the teaching cost of Algorithm 1 can still be bounded well, and it is robust to small misspecification levels. The results are stated in the following theorem.

**Theorem 3** *For $\epsilon$-approximate teachable epiphany learners as defined in Definition 2, if $\|\boldsymbol{\theta}_0 - \boldsymbol{\theta}^\star\|_2 \le \delta_0$, the cost function is bounded from above by $c_{max}$, the confidence set $\mathcal{C}_t$ is constructed according to Lemma 1 with $C = \mathcal{O}(\epsilon\gamma c_{max}/(1-\gamma)^2 + C_\star)$, and if $\lambda = 1/\delta_0^2$, then with probability at least $1 - 3\delta$, the teaching cost incurred by running Algorithm 1 is upper bounded by*

$$\mathcal{O}\left(C \cdot \left(1 + d\sqrt{\log\left(1 + \frac{C^2\delta_0^2\log\delta}{\log\gamma}\right)}\right) \cdot \sqrt{\frac{\log\delta}{\log\gamma}}\log\left(\frac{C\log\delta}{\delta\log\gamma}\right) \cdot \frac{\epsilon\log\delta}{\log\gamma}C\right). \tag{9}$$

In contrast to Theorem 2, the major difference is that there is one extra cost term in Theorem 3 due to the intrinsic bias of the linear approximation. When $\epsilon$ is sufficiently small, those terms with coefficient $\epsilon$ can be ignored safely, which gives us the following proposition.

**Proposition 1** *Under the same assumptions as Theorem 3, if $\epsilon = \mathcal{O}\left(C_\star(1-\gamma)^2/(\gamma c_{max})\right)$ then with probability at least $1 - 3\delta$, the total cost incurred by running Algorithm 1 is upper bounded by*

$$\mathcal{O}\left(C_\star \cdot \left(1 + d\sqrt{\log\left(1 + \frac{C_\star^2\delta_0^2\log\delta}{\log\gamma}\right)}\right) \cdot \sqrt{\frac{\log\delta}{\log\gamma}}\log\left(\frac{C_\star\log\delta}{\delta\log\gamma}\right) + \frac{\epsilon\log\delta}{\log\gamma}C_\star\right).$$

Hence, as indicated by Theorem 3 and Proposition 1, our algorithm can still attain good theoretical guarantees when the misspecification level is low.

# 6  A Numerical Case Study

In this section, we provide a case study on a synthetic learner to illustrate the algorithm. We also evaluate how the choice of $\lambda$ affects the empirical teaching cost, as $\lambda$ plays a critical role in our algorithm design.

## 6.1  Experimental Setup

**Knowledge states and teaching instructions**. We sample 100 weights $\{\boldsymbol{h}_i\}_{i=1}^{100}$ uniformly at random from $[-3,3]^d$ to simulate different knowledge states, each of which corresponds to a linear regressor. We then pick one of the weights to represent the target knowledge state, denoted as $\boldsymbol{h}^\star$. To generate the teaching instructions, we first sample 20 points $\{\boldsymbol{z}_i\}_{i=1}^{20}$ from a normal distribution $\mathcal{N}(\boldsymbol{0}, \boldsymbol{I})$, and their corresponding labels are generated by $y_i = \langle \boldsymbol{h}^\star, \boldsymbol{z}_i \rangle + \zeta$, where $\zeta \sim \mathcal{N}(0,1)$ is the observation noise. By $\{\boldsymbol{x}_i\}_{i=1}^{20}$ we denote the set of teaching instructions, where $\boldsymbol{x}_i = (\boldsymbol{z}_i, y_i)$.

**Feature representation**. We consider the feature representation for each triplet $(\boldsymbol{h}, \boldsymbol{x}, \boldsymbol{h}')$ to be

$$\boldsymbol{\phi}(\boldsymbol{h}'|\boldsymbol{h}, \boldsymbol{x}) = \left[1/\left(Z_{(\boldsymbol{h},\boldsymbol{x})}^{(1)} \cdot \|\boldsymbol{h}' - \boldsymbol{h} + \eta\nabla_{\boldsymbol{h}}\ell(\boldsymbol{h}, \boldsymbol{x})\|_2\right), \quad 1/\left(Z_{(\boldsymbol{h},\boldsymbol{x})}^{(2)} \cdot \|\nabla_{\boldsymbol{h}}\ell(\boldsymbol{h}, \boldsymbol{x})\|_2\right)\right] \tag{10}$$

where $\eta$ is the learning rate, and $Z_{(\boldsymbol{h},\boldsymbol{x})}^{(i)}$ is the normalizing constant for the $i^{th}$ dimension of the feature representation $\boldsymbol{\phi}(\cdot|\boldsymbol{h}, \boldsymbol{x})$. The normalizing constants are used to ensure that $\sum_{\boldsymbol{h}'\in\mathcal{H}}\boldsymbol{\phi}(\boldsymbol{h}'|\boldsymbol{h}, \boldsymbol{x}) = (1,1)$. Therefore, all the feasible $\boldsymbol{\theta}$ that forms a probabilistic distribution lies in a 1-$d$ simplex. Intuitively, the first dimension indicates that the learner is more likely to transit to those knowledge states that align well with the updated knowledge state, i.e., $\boldsymbol{h} - \eta\nabla_{\boldsymbol{h}}\ell(\boldsymbol{h}, \boldsymbol{x})$, whereas the second dimension implies that the learner's knowledge state transition will become more random if the teaching instruction is more difficult, which is measured by the gradient norm $\|\nabla_{\boldsymbol{h}}\ell(\boldsymbol{h}, \boldsymbol{x})\|_2$.

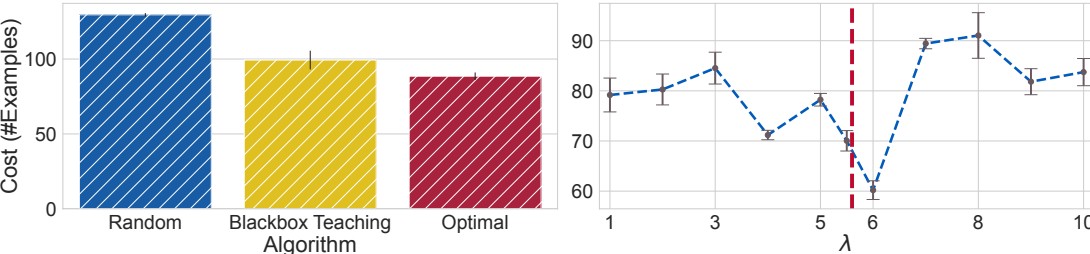

Figure 4: Left: A comparison between random teaching policy, black-box teaching policy and the optimal teaching policy in terms of the mean of the averaged teaching cost for 99 initial states; Right: Effect of different values of $\lambda$ on the teaching cost (computed with the first 10 states to save computation time).

### 6.2 Empirical Results

**Comparing with baselines.** We first evaluate the empirical performance of Algorithm 1 under the above experimental setup. Specifically, we set the learning rate $\eta = 1$, and compare it with the random teaching policy and the optimal policy. We compute the mean of the averaged teaching cost of starting from each non-goal knowledge state (99 states). The averaged teaching cost is computed with 50 random seeds. The results are presented in figure 4 left. As expected, the black-box teaching algorithm outperforms the random teaching policy but underperforms the optimal teaching policy.

**How to set $\lambda$?** The initialization plays an important role in our algorithm design and theoretical analysis. Our theoretical analysis has demonstrated the impact of the initialization on the teaching cost. However, given the initialization, it is still unclear how to set the right regularization parameter $\lambda$. We conjecture that the 'optimal' $\lambda$ should be around $1/\delta_0^2$, which is also adopted in our theoretical analysis. To verify this idea, we study how the choice of $\lambda$ affects the teaching cost. Under the same setting as above, we vary the value of $\lambda$ in $\{1.0, 1.5, ..., 10.0\}$. To save computation time, we adopt the first 10 states to serve as the initial state and repeat the previous experiments. The results are reported in the right plot of figure 4. The red dashed line corresponds to the line of $x = 1/\delta_0^2$ with $\delta_0^2 = 0.18$. Based on the empirical results, we can observe that the best choice of $\lambda$ is 6, which is close to $1/\delta_0^2$. In addition, if we set $\lambda$ too large or too small, the teaching cost will increase accordingly.

In summary, our experimental results highlight that modelling the learner's learning dynamics is crucial to achieve a low teaching cost. Furthermore, given the initialization, setting $\lambda = 1/\delta_0^2$ is a reasonable choice for obtaining good empirical performance.

## 7 Conclusion

In this paper, we investigate a generic framework for machine teaching, under which the learner's dynamics can be represented as an MDP with unknown, learnable parameters. To solve the teaching problem, we introduce an algorithm that accommodates both epiphany and non-epiphany learners, thus bridging a significant gap in the current literature. Moreover, we furnish a rigorous analysis of the teaching costs associated with these two types of learners under disparate settings. Complementing our theoretical insights, we conduct empirical research to demonstrate the efficiency of our proposed algorithm and provide a guideline for setting hyperparameters. It is our aspiration that this work will stimulate future research in proposing more nuanced assumptions about the structure of the learner's MDP and more efficient algorithms for machine teaching.

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

## A    Appendix

In the appendix, we present the proofs of our theorems. The proofs of Lemma 1, Theorem 1, Theorem 2 and Theorem 3 can be found in sections D, E, F and G, respectively. In section H, we provide a reference to the existing lemmas that we rely on.

## B    Additional Discussions

**Limitations.**  First of all, our algorithm requires the knowledge of the upper bound of $C_\star$ and $\delta_0$ for setting the hyperparameters. To address this issue, we can use binary search to find a good choice of these hyperparameters in practice. Secondly, it assumes a pregiven and *fixed* feature mapping during the entire teaching process, which might be suboptimal in reality and lead to biased teaching policies. Lastly, we assume that the teaching algorithm can observe the learner's state at every iteration, which might be expensive for learners such as humans (i.e., one needs to quiz the learner at every iteration).

**Future directions.**  As future directions, it would be exciting to incorporate representation learning into the entire framework. Besides, instead of assuming the learner's state to be observable, a partially observable setting could be considered, under which probing the learner's state will incur some cost. Therefore, the algorithm should smartly determine when probing the learner's state is necessary. In addition, it would be interesting to propose more fine-grained and learner-specific structural assumptions on the underlying MDPs. For example, for version space learners, the knowledge states will form a topological order; for forgetful learners, there will be a non-zero probability to transit back to the *previous* knowledge states in the topological graph. By exploiting the structural assumptions, we will design more efficient algorithms and achieve stronger theoretical guarantees. Lastly, it would be interesting to incorporate ideas from meta learning for learning the initialization $\boldsymbol{\theta}_0$.

## C    Extended Backgrounds on Various Learner's Models

**Version-space learner.**    The version space learner was studied by Goldman and Kearns (1995) for machine teaching. The hypothesis class of the version space learner is usually a finite set $\mathcal{H}$, which contains a target hypothesis $h^\star \in \mathcal{H}$. The teacher can pick a teaching example from the ground set $\mathcal{X}$ to teach the learner. Once an example $(x, h^\star(x)))$ is provided to the learner, the learner will update its version space by removing those hypotheses that are not consistent with the example, i.e., $\mathcal{H} \leftarrow \mathcal{H} \setminus \{h \in \mathcal{H} | h(x) \neq h^\star(x)\}$. Under this teacher-learner interaction protocol, the teacher *knows* the aforementioned update rule of the learner. The entire problem is essentially a set cover problem, which is NP-hard. But a greedy-approximation algorithm admits a teaching complexity of $\mathcal{O}(\log(\mathcal{H}) \cdot C_\star)$, where $C_\star$ is the optimal teaching cost. The version space learner can also be regarded as a tabular case of the machine teaching problem, which falls in the Markov learner case, i.e., a special case of our teaching framework.

**Black-box version-space learner.**    The black-box version-space learner was studied in Dasgupta et al. (2019). In this framework, they assume the teacher does not know the hypothesis class $\mathcal{H}$ at the beginning, but the teacher knows the learner's dynamics rule (i.e., how does the learner update the knowledge state). Then the teaching problem is equivalent to the *online set cover* problem. The analysis of the online set cover applies to the analysis of the teaching cost. This work can be regarded as a complement to our work, as they assume the learner's model is known, but the state is unknown. Our work assumes the learner's state is observable, but the learner's model is unknown.

**Black-box iterative learner.**    The black-box iterative learner Liu et al. (2017) is in the same philosophy as Dasgupta et al. (2019). The main difference is that, for the black-box iterative learner, it deals with gradient-based learner, i.e., the learner updates it by following the gradient descent rule. Therefore, this work still assumes the learner's model is known.

**Memory-based learner.**    The memory-based learner was studied in Settles and Meeder (2016); Hunziker et al. (2019) for modeling the forgetting behavior of human learning. In these works, they used the half-life

model as a proxy to model the human learner's model. Specifically, in Hunziker et al. (2019) the teaching problem was formulated as a submodular maximization problem (maximizing the memorization utility of the underlying learner) due to the property of the half-life model.

**Bayesian knowledge tracing (BKT) learner.** As an instance of skill-based learners (Whitehill and Movellan, 2017), BKT assumes that student knowledge is represented as a set of binary variables, one per skill, where the skill is either mastered by the student or not. Observations in BKT are also binary: a student gets a problem/step either right or wrong. The learner's state is updated by Bayes rule given the new observation. Hence, the teacher still knows the learner's model.

## D   Proof of Lemma 1

**Lemma 1** *Under Assumptions 1 and 2, for any $t \geq 1$, with probability at least $1 - \delta$, we have that the true parameter $\boldsymbol{\theta}^\ell$ lies in*

$$\mathcal{C}_t = \left\{ \boldsymbol{\theta} \in \mathbb{R}^d \,\middle|\, \|\hat{\boldsymbol{\theta}}_t - \boldsymbol{\theta}\|_{\boldsymbol{\Sigma}_t} \leq C\sqrt{d \log\left((4(t^2 + t^3 C^2/\lambda))/\delta\right)} + \sqrt{\lambda}\delta_0 \right\}. \tag{6}$$

*Proof:* We prove this by induction on $k$, which is the index of the value functions returned by EVI. By definition, the fitted value function in the interval $[t_k, t_{k+1} - 1]$ is $V_k(\cdot)$. To be noted, since when $t = 0$, we must have $\boldsymbol{\theta}^\ell \in \mathcal{C}_0$ by Assumption 2. Therefore, we abuse the notation a little bit by reloading $t_0 = 1$ for the proof. Therefore, we first prove the base step, where $t \in [1, t_1 - 1]$. For notation simplicity, we define

$$\boldsymbol{\phi}_m = \boldsymbol{\phi}_{V_0}(\boldsymbol{h}_m, \boldsymbol{x}_m), \ \boldsymbol{\Phi}_t = (\boldsymbol{\phi}_1, ...., \boldsymbol{\phi}_t), \ \boldsymbol{v}_t = (V_0(\boldsymbol{h}_2), ..., V_0(\boldsymbol{h}_{t+1}))^\top.$$

Recall the definition of $\hat{\boldsymbol{\theta}}_t$, by rewriting it in the matrix form, we get

$$\begin{aligned}
\hat{\boldsymbol{\theta}}_t &= \boldsymbol{\Sigma}_t^{-1} \boldsymbol{b}_t = \boldsymbol{\Sigma}_t^{-1} \left( \lambda \boldsymbol{\theta}_0 + \sum_{m=1}^t \boldsymbol{\phi}_m V_0(\boldsymbol{h}_{m+1}) \right) \\
&= \left( \lambda \boldsymbol{I} + \boldsymbol{\Phi}_t \boldsymbol{\Phi}_t^\top \right)^{-1} (\lambda \boldsymbol{\theta}_0 + \boldsymbol{\Phi}_t \boldsymbol{v}_t) \\
&= \left( \lambda \boldsymbol{I} + \boldsymbol{\Phi}_t \boldsymbol{\Phi}_t^\top \right)^{-1} \boldsymbol{\Phi}_t (\boldsymbol{v}_t - \boldsymbol{\Phi}_t^\top \boldsymbol{\theta}_0) + \boldsymbol{\theta}_0. \\
&= \boldsymbol{\Sigma}_t^{-1} \boldsymbol{\Phi}_t (\boldsymbol{v}_t - \boldsymbol{\Phi}_t^\top \boldsymbol{\theta}_0) + \boldsymbol{\theta}_0.
\end{aligned}$$

Next, we define the following random variables

$$\eta_m = V_0(s_{m+1}) - \langle \boldsymbol{\phi}_m, \boldsymbol{\theta}^\ell \rangle, \quad \boldsymbol{\eta}_t = (\eta_1, ..., \eta_t)^\top.$$

Since $C \geq C_\star$, the sequence $\{\eta_t\}_{t=1}^{t_1}$ are $C$-sub-Gaussian. Now, we can rewrite $\hat{\boldsymbol{\theta}}_t$ as

$$\begin{aligned}
\hat{\boldsymbol{\theta}}_t &= \boldsymbol{\Sigma}_t^{-1} \boldsymbol{\Phi}_t \left( \boldsymbol{\eta}_t + \boldsymbol{\Phi}_t^\top (\boldsymbol{\theta}^\ell - \boldsymbol{\theta}_0) \right) + \boldsymbol{\theta}_0 \\
&= \boldsymbol{\Sigma}_t^{-1} \boldsymbol{\Phi}_t \boldsymbol{\eta}_t + \boldsymbol{\Sigma}_t^{-1} \boldsymbol{\Phi}_t \boldsymbol{\Phi}_t^\top (\boldsymbol{\theta}^\ell - \boldsymbol{\theta}_0) + \boldsymbol{\theta}_0.
\end{aligned}$$

By subtracting $\boldsymbol{\theta}^\ell$ on both sides, we get

$$\begin{aligned}
\hat{\boldsymbol{\theta}}_t - \boldsymbol{\theta}^\ell &= \boldsymbol{\Sigma}_t^{-1} \boldsymbol{\Phi}_t \boldsymbol{\eta}_t + \left( \boldsymbol{\Sigma}_t^{-1} \boldsymbol{\Phi}_t \boldsymbol{\Phi}_t^\top - \boldsymbol{I} \right) (\boldsymbol{\theta}^\ell - \boldsymbol{\theta}_0) \\
&= \boldsymbol{\Sigma}_t^{-1} \boldsymbol{\Phi}_t \boldsymbol{\eta}_t + \boldsymbol{\Sigma}_t^{-1} \left( \boldsymbol{\Phi}_t \boldsymbol{\Phi}_t^\top - \boldsymbol{\Sigma}_t \right) (\boldsymbol{\theta}^\ell - \boldsymbol{\theta}_0) \\
&= \boldsymbol{\Sigma}_t^{-1} \boldsymbol{\Phi}_t \boldsymbol{\eta}_t + \lambda \boldsymbol{\Sigma}_t^{-1} (\boldsymbol{\theta}_0 - \boldsymbol{\theta}^\ell).
\end{aligned}$$

Then, we further obtain the following by the Cauchy-Schwarz inequality,

$$\begin{aligned}
\left\| \hat{\boldsymbol{\theta}}_t - \boldsymbol{\theta}^\ell \right\|_{\boldsymbol{\Sigma}_t}^2 &= \left\langle \boldsymbol{\Sigma}_t(\hat{\boldsymbol{\theta}}_t - \boldsymbol{\theta}^\ell), \boldsymbol{\Phi}_t \boldsymbol{\eta}_t \right\rangle_{\boldsymbol{\Sigma}_t^{-1}} + \lambda \left\langle \boldsymbol{\Sigma}_t(\hat{\boldsymbol{\theta}}_t - \boldsymbol{\theta}^\ell), \boldsymbol{\theta}_0 - \boldsymbol{\theta}^\ell \right\rangle_{\boldsymbol{\Sigma}_t^{-1}} \\
&\leq \left\| \boldsymbol{\Sigma}_t(\hat{\boldsymbol{\theta}}_t - \boldsymbol{\theta}^\ell) \right\|_{\boldsymbol{\Sigma}_t^{-1}} \left( \|\boldsymbol{\Phi}_t \boldsymbol{\eta}_t\|_{\boldsymbol{\Sigma}_t^{-1}} + \lambda \|\boldsymbol{\theta}_0 - \boldsymbol{\theta}^\ell\|_{\boldsymbol{\Sigma}_t^{-1}} \right) \\
&= \left\| \hat{\boldsymbol{\theta}}_t - \boldsymbol{\theta}^\ell \right\|_{\boldsymbol{\Sigma}_t} \left( \|\boldsymbol{\Phi}_t \boldsymbol{\eta}_t\|_{\boldsymbol{\Sigma}_t^{-1}} + \lambda \|\boldsymbol{\theta}_0 - \boldsymbol{\theta}^\ell\|_{\boldsymbol{\Sigma}_t^{-1}} \right).
\end{aligned}$$

By Lemma 6 from Abbasi-Yadkori et al. (2011), for any $t \in [1, t_1]$, we have the following hold with probability at least $1 - \delta/(t_1(t_1 + 1))$,

$$
\begin{aligned}
\|\boldsymbol{\Phi}_t \boldsymbol{\eta}_t\|_{\boldsymbol{\Sigma}_t^{-1}} &\leq C \sqrt{2 \log \left( \frac{\det(\boldsymbol{\Sigma}_t)^{1/2}}{\lambda^{d/2} \cdot \delta/(t_1(t_1 + 1))} \right)} \\
&\leq C \sqrt{2 \log \left( \frac{(\lambda + tC^2)^{d/2}}{\lambda^{d/2} \cdot \delta/(t_1(t_1 + 1))} \right)} \\
&\leq C \sqrt{d \log \left( \frac{1 + tC^2/\lambda}{\delta/(t_1(t_1 + 1))} \right)} \\
&= C \sqrt{d \log \left( \frac{t_1(t_1 + 1) + t \cdot t_1(1 + t_1)C^2/\lambda}{\delta} \right)}.
\end{aligned}
$$

In the next, we bound $\|\boldsymbol{\theta}_0 - \boldsymbol{\theta}^\ell\|_{\boldsymbol{\Sigma}_t^{-1}}$,

$$
\left\| \boldsymbol{\theta}_0 - \boldsymbol{\theta}^\ell \right\|_{\boldsymbol{\Sigma}_t^{-1}}^2 \leq \frac{1}{\lambda_{\min}(\boldsymbol{\Sigma}_t)} \|\boldsymbol{\theta}_0 - \boldsymbol{\theta}^\ell\|_2^2 = \frac{1}{\lambda} \|\boldsymbol{\theta}_0 - \boldsymbol{\theta}^\ell\|_2^2.
$$

Finally, by plugging in the above bounds, we get the desired result for the base step

$$
\left\| \hat{\boldsymbol{\theta}}_t - \boldsymbol{\theta}^\ell \right\|_{\boldsymbol{\Sigma}_t} \leq C \sqrt{d \log \left( \frac{t_1(t_1 + 1) + t \cdot t_1(1 + t_1)C^2/\lambda}{\delta} \right)} + \sqrt{\lambda} \|\boldsymbol{\theta}_0 - \boldsymbol{\theta}^\ell\|_2.
$$

Since $2t \geq t_1$, then we have

$$
\left\| \hat{\boldsymbol{\theta}}_t - \boldsymbol{\theta}^\ell \right\|_{\boldsymbol{\Sigma}_t} \leq C \sqrt{d \log \left( \frac{4(t^2 + t^3C^2)/\lambda)}{\delta} \right)} + \sqrt{\lambda} \delta_0. \tag{11}
$$

Let's suppose that, for any $k \in \{0, ..., n-1\}$, equation 11 holds for all $t \in [t_k, t_{k+1} - 1]$. For the induction step, we define the following notations for any $k \in \{0, ..., n-1\}$,

$$
\breve{V}_k(\cdot) = \min \{ C, V_k(\cdot) \}.
$$

Consequently, for any $k \in \{0, ..., n\}$ and $t \in [t_k, t_{k+1} - 1]$, we further define

$$
\breve{\boldsymbol{\Sigma}}_t = \lambda \boldsymbol{I} + \sum_{i=1}^{t} \boldsymbol{\phi}_{\breve{V}_{k(i)}}(\boldsymbol{h}_i, \boldsymbol{x}_i) \boldsymbol{\phi}_{\breve{V}_{k(i)}}(\boldsymbol{h}_i, \boldsymbol{x}_i)^\top, \quad \breve{\boldsymbol{\mu}}_t = \lambda \boldsymbol{\theta}_0 + \sum_{i=1}^{t} \boldsymbol{\phi}_{\breve{V}_{k(i)}}(\boldsymbol{h}_i, \boldsymbol{x}_i) \breve{V}_{k(i)}(\boldsymbol{h}_{i+1}),
$$

In analogy, we reload the definition for $\hat{\boldsymbol{\theta}}_t$ and $\eta_t$ by

$$
\breve{\boldsymbol{\theta}}_t = \breve{\boldsymbol{\Sigma}}_t^{-1} \breve{\boldsymbol{\mu}}_t, \quad \eta_t = \breve{V}_{k(t)}(\boldsymbol{h}_{t+1}) - \langle \boldsymbol{\phi}_{\breve{V}_{k(t)}}(\boldsymbol{h}_t, \boldsymbol{x}_t), \boldsymbol{\theta}^\ell \rangle
$$

By the above definition, it's easy to verify that $\{\breve{\eta}_t\}_{t=1}^{t_n}$ is almost surely $C$-sub-Gaussian.[3] Then, we can apply the Lemma 6 again, and conclude that $\boldsymbol{\theta}^\ell \in \breve{\mathcal{C}}_t$ holds with probability at least $1 - \delta/(t_n(t_n + 1))$ for any $t \in [t_n, t_{n+1} - 1]$ with

$$
\breve{\mathcal{C}}_t = \left\{ \boldsymbol{\theta} \in \mathbb{R}^d \,\Big|\, \|\breve{\boldsymbol{\theta}}_t - \boldsymbol{\theta}\|_{\breve{\boldsymbol{\Sigma}}_t} \leq C \sqrt{d \log \left( \frac{4(t^2 + t^3C^2/\lambda)}{\delta} \right)} + \sqrt{\lambda} \delta_0 \right\}.
$$

By the optimism principle in Algorithm 2 and the base step of induction, we will have $\breve{V}_k(\cdot) = \breve{V}_k$ for $k \in \{0, ..., n-1\}$, which further gives us that $\breve{\boldsymbol{\Sigma}}_t = \boldsymbol{\Sigma}_t$, $\breve{\boldsymbol{\mu}}_t = \boldsymbol{\mu}_t$, $\breve{\eta}_t = \eta_t$ and $\breve{\boldsymbol{\theta}}_t = \hat{\boldsymbol{\theta}}_t$ for all $t \in [1, t_{n+1} - 1]$.

---

[3]To be noted, without such construction, if the induction step conditions on the base step, there is no guarantee that the (conditional) distribution of $\eta_t$ is $C$-sub-Gaussian. This may prevent us from applying the Lemma 6.

Consequently, we further have $\check{\mathcal{C}}_t = \mathcal{C}_t$. Lastly, by applying the union bound over $k \geq 0$, we will get that the probability of the event in Lemma 1 holds is at least

$$1 - \sum_{k=0} \frac{\delta}{t_k(t_k+1)} \geq 1 - \delta.$$

$\square$

## E   Proof of Theorem 1

**Theorem 1** *Under Assumptions 1 and 2, if the confidence set $\mathcal{C}_t$ is constructed according to Lemma 1 with $C = \mathcal{O}(C_\star)$, $\lambda = 1/\delta_0^2$, and the cost function is bounded from below by $c_{\min}$ for all non-goal knowledge states $(\mathcal{H} \setminus \{h^\star\})$ and teaching instruction $(\mathcal{X})$ pairs, then with probability at least $1 - 2\delta$, the teaching cost of Algorithm 1 for non-epiphany learners (i.e., $\gamma = 1$) is upper bounded by*

$$\mathcal{O}\left(\left(1 + d\sqrt{\log\left(1 + \frac{C_\star d\delta_0}{\delta c_{\min}}\right)}\right) \cdot \log^{1.5}\left(\frac{C_\star d}{c_{\min}\delta}\right) \cdot \frac{C_\star^2 d}{c_{\min}}\right). \tag{7}$$

*Proof:* To prove Theorem 1, we first bound the teaching cost for running Algorithm 1 for $T$ steps. Then, we can derive a bound for $T$, and plugging it back to obtain the final result.

For any $T$, we can decompose the teaching cost into the following

$$\sum_{t=0}^{T} c(\boldsymbol{h}_t, \boldsymbol{x}_t) \leq \sum_{t=0}^{T} c(\boldsymbol{h}_t, \boldsymbol{x}_t) - V_0(\boldsymbol{h}_0) + C. \tag{12}$$

By Lemma 3, we know that

$$-\sum_{t=1}^{T} \left(V_{k(t)}(\boldsymbol{h}_t) - V_{k(t)}(\boldsymbol{h}_{t+1})\right) + 2dC\log\left(1 + \frac{TC^2}{\lambda}\right) + C\log\left(1 + \frac{2T}{\lambda}\right) + V_0(\boldsymbol{h}_0) \geq 0.$$

By adding it to the r.h.s of equation 12, we get

$$\sum_{t=0}^{T} c(\boldsymbol{h}_t, \boldsymbol{x}_t) \leq \sum_{t=0}^{T} c(\boldsymbol{h}_t, \boldsymbol{x}_t) - \cancel{V_0(\boldsymbol{h}_0)} + \sum_{t=1}^{T} \left(V_{k(t)}(\boldsymbol{h}_{t+1}) - V_{k(t)}(\boldsymbol{h}_t)\right)$$
$$+ 2dC\log\left(1 + \frac{TC^2}{\lambda}\right) + C\log\left(1 + \frac{2T}{\lambda}\right) + \cancel{V_0(\boldsymbol{h}_0)} + C.$$

By rearranging the above terms, we can get the following terms

$$\sum_{t=0}^{T} c(\boldsymbol{h}_t, \boldsymbol{x}_t) \leq \underbrace{\sum_{t=0}^{T} \left[c(\boldsymbol{h}_t, \boldsymbol{x}_t) + \mathbb{P}_{\boldsymbol{\theta}^\ell} V_{k(t)}(\boldsymbol{h}_t, \boldsymbol{x}_t) - V_{k(t)}(\boldsymbol{h}_t)\right]}_{\textcircled{1}} + \underbrace{\sum_{t=0}^{T} \left[V_{k(t)}(\boldsymbol{h}_{t+1}) - \mathbb{P}_{\boldsymbol{\theta}^\ell} V_{k(t)}(\boldsymbol{h}_t, \boldsymbol{x}_t)\right]}_{\textcircled{2}}$$
$$+ 2dC\log\left(1 + \frac{TC^2}{\lambda}\right) + C\log\left(1 + \frac{2T}{\lambda}\right) + C.$$

In the next, it remains to bound $\textcircled{1}$ and $\textcircled{2}$. By Lemma 4, we can bound $\textcircled{1}$ by

$$\textcircled{1} \leq 4\beta_T \sqrt{2Td \cdot \log\left(1 + \frac{TC^2}{\lambda}\right)} + 2(C+1) \cdot \left(2d\log\left(1 + \frac{TC^2}{\lambda}\right) + \log\left(1 + \frac{T}{\lambda}\right) + 1\right).$$

Then, by Lemma 9, we can bound the martingale difference ②, with probability at least $1 - \delta$, by

$$② \leq 2C\sqrt{2T\log\left(\frac{T}{\delta}\right)}.$$

By merging the terms, we simplify the upper bound of the teaching cost to be

$$\sum_{t=1}^{T} c(\boldsymbol{h}_t, \boldsymbol{x}_t) \leq 4\beta_T\sqrt{2Td \cdot \log\left(1 + \frac{TC^2}{\lambda}\right)} + 15Cd\log\left(1 + \frac{TC^2}{\lambda}\right) + 2C\sqrt{2T\log\left(\frac{T}{\delta}\right)},$$

where $\beta_T = C\sqrt{d\log\left((4(T^2 + T^3C^2/\lambda))/\delta\right)} + \sqrt{\lambda}\delta_0$. Now, it remains to bound $T$. Since the cost function is bounded from below by $c_{\min}$, then we will have

$$T \cdot c_{\min} \leq \sum_{t=0}^{T} c(\boldsymbol{h}_t, \boldsymbol{x}_t).$$

By replacing the r.h.s. term with the upper bound derived, we get

$$T \cdot c_{\min} - C \leq 4\beta_T\sqrt{2Td \cdot \log\left(1 + \frac{TC^2}{\lambda}\right)} + 15Cd\log\left(1 + \frac{TC^2}{\lambda}\right) + 2C\sqrt{2T\log\left(\frac{T}{\delta}\right)}.$$

For the terms on the r.h.s, we can loosely bound the second term by

$$15Cd\log\left(1 + \frac{TC^2}{\lambda}\right) \leq 8\beta_T\sqrt{2Td \cdot \log\left(1 + \frac{TC^2}{\lambda}\right)}.$$

Then, we can bound $T$ by

$$T \leq \frac{1}{c_{\min}}\left(12\beta_T\sqrt{2d\log\left(1 + \frac{TC^2}{\lambda}\right)} + 2C\sqrt{2\log\left(\frac{T}{\delta}\right)}\right) \cdot \sqrt{T} + \frac{C}{c_{\min}}.$$

Using the fact that $c \leq a\sqrt{c} + b \Rightarrow c \leq (a + \sqrt{b})^2$ for $a, b \geq 0$, we have

$$T \leq \left(\frac{1}{c_{\min}^2}\left(12\beta_T\sqrt{2d\log\left(1 + \frac{TC^2}{\lambda}\right)} + 2C\sqrt{2\log\left(\frac{T}{\delta}\right)}\right) + \sqrt{\frac{C}{c_{\min}}}\right)^2.$$

By using the inequality $(a + b)^2 \leq 2a^2 + 2b^2$ twice, we get

$$T \leq \frac{32}{c_{\min}^2}\left(36\beta_T^2 d\log\left(1 + \frac{TC^2}{\lambda}\right) + C^2\log\left(\frac{T}{\delta}\right)\right) + \frac{2C}{c_{\min}}.$$

Plugging in the following upper bound of $\beta_T^2$,

$$\beta_T^2 \leq 2C^2 d\log\left(\frac{4(T^2 + T^3C^2/\lambda)}{\delta}\right) + 2\lambda\delta_0^2 d,$$

Then, we get

$$T \leq \frac{32}{c_{\min}^2}\left(72\left(C^2 d^2\log\left(\frac{4T^2 + 4T^3C^2/\lambda}{\delta}\right) + \lambda\delta_0^2 d^2\right) \cdot \log\left(1 + \frac{TC^2}{\lambda}\right) + C^2\log\left(\frac{T}{\delta}\right)\right) + \frac{2C}{c_{\min}}.$$

By rearranging the terms, we get

$$T \leq \frac{2304 C^2 d^2}{c_{\min}^2} \cdot \log\left(\frac{4T^2 + 4T^3 C^2/\lambda}{\delta}\right) \cdot \log\left(1 + \frac{TC^2}{\lambda}\right)$$
$$+ \frac{2304 \lambda d^2 C^2 \delta_0^2}{c_{\min}^2} \cdot \log\left(1 + \frac{TC^2}{\lambda}\right)$$
$$+ \frac{32 C^2}{c_{\min}^2} \cdot \log\left(\frac{T}{\delta}\right) + \frac{2C}{c_{\min}}.$$

Since $\lambda = 1/\delta_0^2$, we can get the following bound

$$T \leq \frac{4608 C^2 d^2}{c_{\min}^2} \cdot \log\left(\frac{4T^2 + 4T^3 C^2 \delta_0^2}{\delta}\right) \cdot \log\left(1 + TC^2\delta_0^2\right)$$
$$+ \frac{32 C^2}{c_{\min}^2} \cdot \log\left(\frac{T}{\delta}\right) + \frac{2C}{c_{\min}}.$$

We now consider the following cases: when $\delta_0 \leq 1/(TC^2)$, we will have, for some universal constant $C_0$,

$$T \leq C_0 \left(\frac{C^2 d^2}{c_{\min}^2} \log^2\left(\frac{T}{\delta}\right)\right).$$

When $\delta_0 > 1/(TC^2)$, we will have, for some universal constant $C_1$,

$$T \leq C_1 \left(\frac{C^2 d^2}{c_{\min}^2} \log^2\left(\frac{TC}{\delta}\right)\right).$$

According to Lemma 5, we arrive at the desired bound for $T$

$$T = \mathcal{O}\left(\frac{C^2 d^2}{c_{\min}^2} \log^2\left(\frac{Cd}{c_{\min}\delta}\right)\right).$$

Because $C = \mathcal{O}(C_\star)$ and plugging in the bound for $T$ into the original bound, we can finally get the desired bound for the teaching cost hold with probability at least $1 - 2\delta$ by further applying union bound on the two events (i.e., Lemma 1 and bounding ②),

$$\sum_t c(\boldsymbol{h}_t, \boldsymbol{x}_t) = \mathcal{O}\left(\left(1 + d\sqrt{\log\left(1 + \frac{C_\star d\delta_0}{\delta c_{\min}}\right)}\right) \cdot \log^{1.5}\left(\frac{C_\star d}{c_{\min}\delta}\right) \cdot \frac{C_\star^2 d}{c_{\min}}\right).$$

$\square$

**Lemma 2** *Under the same assumptions as Theorem 1, if Algorithm 1 runs for $T$ steps, then the total number of value function updates (i.e., the number of EVI calls) $K$ is at most*

$$K \leq 2d \log\left(1 + \frac{TC^2}{\lambda}\right) + \log\left(1 + \frac{2T}{\lambda}\right).$$

*Proof:* The value function update can be triggered by either the determinant criteria ($K_1$) or the iteration criteria ($K_2$). We bound each part separately.

**Bounding $K_1$:** To bound $K_1$, it suffices to bound the determinant of $\boldsymbol{\Sigma}_T$. By Lemma 7, the fact that $\boldsymbol{\Sigma}_0 = \lambda \boldsymbol{I}$, and the Assumption 1, we have

$$\det(\boldsymbol{\Sigma}_T) \leq (\lambda + TC^2)^d.$$

Therefore, we can immediately bound $K_1$ by

$$2^{K_1} \cdot \det(\boldsymbol{\Sigma}_0) = 2^{K_1} \cdot \lambda^d \leq (\lambda + TC^2)^d$$

$$\Rightarrow \quad K_1 \leq 2d \log\left(1 + \frac{TC^2}{\lambda}\right).$$

**Bounding $K_2$:** To bound $K_2$, we can look at the criteria triggered by it, which immediately gives us that

$$(1 + \lambda) \cdot 2^{K_2} \leq T + \lambda$$

$$\Rightarrow \quad K_2 \leq \log\left(\frac{T + \lambda}{1 + \lambda}\right) \leq \log\left(1 + \frac{T}{\lambda}\right).$$

Since $K = K_1 + K_2$, we can conclude that

$$K \leq 2d \log\left(1 + \frac{TC^2}{\lambda}\right) + \log\left(1 + \frac{T}{\lambda}\right).$$

$\square$

**Lemma 3** *Under the same assumptions as Theorem [1], for any $T$, the following holds,*

$$\sum_{t=0}^{T} \left(V_{k(t)}(\boldsymbol{h}_t) - V_{k(t)}(\boldsymbol{h}_{t+1})\right) \leq 2dC \log\left(1 + \frac{TC^2}{\lambda}\right) + C \log\left(1 + \frac{2T}{\lambda}\right) + V_0(\boldsymbol{h}_0).$$

*Proof:* By Lemma [2], we can divide the $T$ steps into $K + 1$ segments, and within each segment, all the steps share the same value function. Let's denote the ending step of $k_{th}$ segment as $t_{k+1} - 1$, then we will have (by canceling out the intermediate terms)

$$\sum_{t=0}^{T} \left(V_{k(t)}(\boldsymbol{h}_t) - V_{k(t)}(\boldsymbol{h}_{t+1})\right) = \sum_{k=0}^{K} V_k(\boldsymbol{h}_{t_k}) - V_k(\boldsymbol{h}_{t_{k+1}}).$$

By rearranging terms, we can further get

$$\sum_{t=0}^{T} \left(V_{k(t)}(\boldsymbol{h}_t) - V_{k(t)}(\boldsymbol{h}_{t+1})\right)$$

$$= \sum_{k=0}^{K-1} \left(V_{k+1}(\boldsymbol{h}_{t_{k+1}}) - V_k(\boldsymbol{h}_{t_{k+1}})\right) + \sum_{k=0}^{K-1} \left(V_k(\boldsymbol{h}_{t_k}) - V_{k+1}(\boldsymbol{h}_{t_{k+1}})\right) + V_K(\boldsymbol{h}_{t_K}) - V_K(\boldsymbol{h}_{t_{K+1}})$$

$$= \sum_{k=0}^{K-1} \left(V_{k+1}(\boldsymbol{h}_{t_{k+1}}) - V_k(\boldsymbol{h}_{t_{k+1}})\right) + V_0(\boldsymbol{h}_{t_0}) - V_K(\boldsymbol{h}_{t_K}) + V_K(\boldsymbol{h}_{t_K}) - V_K(\boldsymbol{h}_{t_{K+1}})$$

$$= \sum_{k=0}^{K-1} \left(V_{k+1}(\boldsymbol{h}_{t_{k+1}}) - V_k(\boldsymbol{h}_{t_{k+1}})\right) + V_0(\boldsymbol{h}_{t_0}) - V_K(\boldsymbol{h}_{t_{K+1}}).$$

Since the value function is non-negative, then we have

$$\sum_{t=1}^{T} \left(V_{k(t)}(\boldsymbol{h}_t) - V_{k(t)}(\boldsymbol{h}_{t+1})\right) \leq K \cdot \max_k \|V_k\|_\infty + V_0(\boldsymbol{h}_0).$$

By plugging in the upper bound of $K$ from Lemma [2] and the upper bound of the value function, $C$, we finally arrive at

$$\sum_{t=1}^{T} \left(V_{k(t)}(\boldsymbol{h}_t) - V_{k(t)}(\boldsymbol{h}_{t+1})\right) \leq 2dC \log\left(1 + \frac{TC^2}{\lambda}\right) + C \log\left(1 + \frac{2T}{\lambda}\right) + V_0(\boldsymbol{h}_0).$$

$\square$

**Lemma 4** *Under the same assumptions as Theorem 1, for any $T$, we can bound ① by,*

$$① = \sum_{t=0}^{T} \left[ c(\boldsymbol{h}_t, \boldsymbol{x}_t) + \mathbb{P}_{\boldsymbol{\theta}^\ell} V_{k(t)}(\boldsymbol{h}_t, \boldsymbol{x}_t) - V_{k(t)}(\boldsymbol{h}_t) \right]$$

$$\leq 4\beta_T \sqrt{2Td \cdot \log\left(1 + \frac{TC^2}{\lambda}\right)} + 2(C+1) \cdot \left( 2d \log\left(1 + \frac{TC^2}{\lambda}\right) + \log\left(1 + \frac{T}{\lambda}\right) + 1 \right),$$

*where $\beta_T = C\sqrt{d \log\left( (4(T^2 + T^3 C^2/\lambda))/\delta) \right)} + \sqrt{\lambda}\delta_0$.*

*Proof:* First of all, by the fact that $V_{k(t)}(\boldsymbol{h}_t) = \min_{\boldsymbol{x} \in \mathcal{X}} Q_{k(t)}(\boldsymbol{h}_t, \boldsymbol{x}) = Q_{k(t)}(\boldsymbol{h}_t, \boldsymbol{x}_t)$, we have

$$① = \sum_{t=0}^{T} \left[ c(\boldsymbol{h}_t, \boldsymbol{x}_t) + \mathbb{P}_{\boldsymbol{\theta}^\ell} V_{k(t)}(\boldsymbol{h}_t, \boldsymbol{x}_t) - Q_{k(t)}(\boldsymbol{h}_t, \boldsymbol{x}_t) \right].$$

Let's suppose that $Q_{k(t)}(\cdot, \cdot)$ is the value function at the $l_{k(t)}$th value iteration of Algorithm 2, i.e., the last iteration of the while loop. Then, based on the EVI algorithm, we have

$$Q_{k(t)}(\boldsymbol{h}_t, \boldsymbol{x}_t) = c(\boldsymbol{h}_t, \boldsymbol{x}_t) + \nu \cdot \min_{\boldsymbol{\theta} \in \mathcal{C}_t \cap \mathcal{B}} \langle \boldsymbol{\theta}, \boldsymbol{\phi}_{V^{(l_{k(t)}-1)}}(\boldsymbol{h}_t, \boldsymbol{x}_t) \rangle$$

$$= c(\boldsymbol{h}_t, \boldsymbol{x}_t) + \nu \cdot \langle \boldsymbol{\theta}_t, \boldsymbol{\phi}_{V^{(l_{k(t)}-1)}}(\boldsymbol{h}_t, \boldsymbol{x}_t) \rangle$$

$$= c(\boldsymbol{h}_t, \boldsymbol{x}_t) + \nu \cdot \langle \boldsymbol{\theta}_t, \boldsymbol{\phi}_{V^{(l_{k(t)})}}(\boldsymbol{h}_t, \boldsymbol{x}_t) \rangle + \nu \cdot \langle \boldsymbol{\theta}_t, [\boldsymbol{\phi}_{V^{(l_{k(t)}-1)}} - \boldsymbol{\phi}_{V^{(l_{k(t)})}}](\boldsymbol{h}_t, \boldsymbol{x}_t) \rangle,$$

where $\boldsymbol{\theta}_t = \arg\min_{\boldsymbol{\theta} \in \mathcal{C}_t \cap \mathcal{B}} \langle \boldsymbol{\theta}, \boldsymbol{\phi}_{V^{(l_{k(t)}-1)}}(\boldsymbol{h}_t, \boldsymbol{x}_t) \rangle$. By plugging the above equation into ① to replace $Q_{k(t)}(\boldsymbol{h}_t, \boldsymbol{x}_t)$, and then rearrange terms, we get

$$c(\boldsymbol{h}_t, \boldsymbol{x}_t) + \mathbb{P}_{\boldsymbol{\theta}^\ell} V_{k(t)}(\boldsymbol{h}_t, \boldsymbol{x}_t) - Q_{k(t)}(\boldsymbol{h}_t, \boldsymbol{x}_t)$$

$$= c(\boldsymbol{h}_t, \boldsymbol{x}_t) + \mathbb{P}_{\boldsymbol{\theta}^\ell} V_{k(t)}(\boldsymbol{h}_t, \boldsymbol{x}_t) - c(\boldsymbol{h}_t, \boldsymbol{x}_t) - \nu \cdot \langle \boldsymbol{\theta}_t, \boldsymbol{\phi}_{V^{(l_{k(t)})}}(\boldsymbol{h}_t, \boldsymbol{x}_t) \rangle$$

$$\quad - \nu \cdot \langle \boldsymbol{\theta}_t, [\boldsymbol{\phi}_{V^{(l_{k(t)}-1)}} - \boldsymbol{\phi}_{V^{(l_{k(t)})}}](\boldsymbol{h}_t, \boldsymbol{x}_t) \rangle$$

$$= \langle \boldsymbol{\theta}^\ell, \boldsymbol{\phi}_{V^{(l_{k(t)})}}(\boldsymbol{h}_t, \boldsymbol{x}_t) \rangle - \nu \cdot \langle \boldsymbol{\theta}_t, \boldsymbol{\phi}_{V^{(l_{k(t)})}}(\boldsymbol{h}_t, \boldsymbol{x}_t) \rangle$$

$$\quad - \nu \cdot \langle \boldsymbol{\theta}_t, [\boldsymbol{\phi}_{V^{(l_{k(t)}-1)}} - \boldsymbol{\phi}_{V^{(l_{k(t)})}}](\boldsymbol{h}_t, \boldsymbol{x}_t) \rangle$$

$$= \langle \boldsymbol{\theta}^\ell - \boldsymbol{\theta}_t, \boldsymbol{\phi}_{V^{(l_{k(t)})}}(\boldsymbol{h}_t, \boldsymbol{x}_t) \rangle + (1 - \nu) \cdot \langle \boldsymbol{\theta}_t, \boldsymbol{\phi}_{V^{(l_{k(t)})}}(\boldsymbol{h}_t, \boldsymbol{x}_t) \rangle$$

$$\quad - \nu \cdot \langle \boldsymbol{\theta}_t, [\boldsymbol{\phi}_{V^{(l_{k(t)}-1)}} - \boldsymbol{\phi}_{V^{(l_{k(t)})}}](\boldsymbol{h}_t, \boldsymbol{x}_t) \rangle.$$

By the termination condition of the EVI algorithm, we have

$$c(\boldsymbol{h}_t, \boldsymbol{x}_t) + \mathbb{P}_{\boldsymbol{\theta}^\ell} V_{k(t)}(\boldsymbol{h}_t, \boldsymbol{x}_t) - Q_{k(t)}(\boldsymbol{h}_t, \boldsymbol{x}_t)$$

$$\leq \langle \boldsymbol{\theta}^\ell - \boldsymbol{\theta}_t, \boldsymbol{\phi}_{V^{(l_{k(t)})}}(\boldsymbol{h}_t, \boldsymbol{x}_t) \rangle + (1 - \nu) \cdot \langle \boldsymbol{\theta}_t, \boldsymbol{\phi}_{V^{(l_{k(t)})}}(\boldsymbol{h}_t, \boldsymbol{x}_t) \rangle + \nu \cdot \frac{1}{\lambda \cdot t'_{k(t)}}$$

$$\leq \langle \boldsymbol{\theta}^\ell - \boldsymbol{\theta}_t, \boldsymbol{\phi}_{V^{(l_{k(t)})}}(\boldsymbol{h}_t, \boldsymbol{x}_t) \rangle + (1 - \nu) \cdot C + \frac{\nu}{\lambda \cdot t'_{k(t)}},$$

where $t'_{k(t)}$ is the time step of $k(t)_{th}$ EVI call, we use $t'_{k(t)}$ instead of $t_{k(t)}$ to avoid ambiguity. Therefore, we can bound ① by

$$① \leq \sum_{t=0}^{T} \langle \boldsymbol{\theta}^\ell - \boldsymbol{\theta}_t, \boldsymbol{\phi}_{V^{(l_{k(t)})}}(\boldsymbol{h}_t, \boldsymbol{x}_t) \rangle + (1 - \nu) \cdot C + \frac{\nu}{\lambda \cdot t'_{k(t)}}$$

$$= \sum_{t=0}^{T} \langle \boldsymbol{\theta}^\ell - \boldsymbol{\theta}_t, \boldsymbol{\phi}_{V^{(l_{k(t)})}}(\boldsymbol{h}_t, \boldsymbol{x}_t) \rangle + \sum_{t=0}^{T} \left( \frac{\nu}{\lambda \cdot t'_{k(t)}} + (1 - \nu) \cdot C \right).$$

By the fact that both $\boldsymbol{\theta}^\ell$ and $\boldsymbol{\theta}_t$ are in $\mathcal{C}_t$ and Lemma 1, we must have

$$\|\boldsymbol{\theta}^\ell - \boldsymbol{\theta}_t\|_{\boldsymbol{\Sigma}_t} \leq 2\beta_t \leq 2\beta_T.$$

Together with the Cauchy-Schwartz inequality, we obtain

$$\begin{aligned}
\langle \boldsymbol{\theta}^\ell - \boldsymbol{\theta}_t, \boldsymbol{\phi}_{V^{(l_{k(t)})}}(\boldsymbol{h}_t, \boldsymbol{x}_t)\rangle &\leq \|\boldsymbol{\theta}^\ell - \boldsymbol{\theta}_t\|_{\boldsymbol{\Sigma}_t} \cdot \|\boldsymbol{\phi}_{V^{(l_{k(t)})}}(\boldsymbol{h}_t, \boldsymbol{x}_t)\|_{\boldsymbol{\Sigma}_t^{-1}} \\
&\leq 2\|\boldsymbol{\theta}^\ell - \boldsymbol{\theta}_t\|_{\boldsymbol{\Sigma}_t} \cdot \|\boldsymbol{\phi}_{V^{(l_{k(t)})}}(\boldsymbol{h}_t, \boldsymbol{x}_t)\|_{\boldsymbol{\Sigma}_t^{-1}} \\
&\leq 4\beta_T \|\boldsymbol{\phi}_{V^{(l_{k(t)})}}(\boldsymbol{h}_t, \boldsymbol{x}_t)\|_{\boldsymbol{\Sigma}_t^{-1}}
\end{aligned}$$

In the meantime, we also have

$$\langle \boldsymbol{\theta}^\ell - \boldsymbol{\theta}_t, \boldsymbol{\phi}_{V^{(l_{k(t)})}}(\boldsymbol{h}_t, \boldsymbol{x}_t)\rangle \leq C.$$

Then, since $C \leq \beta_T$, we get

$$\begin{aligned}
\langle \boldsymbol{\theta}^\ell - \boldsymbol{\theta}_t, \boldsymbol{\phi}_{V^{(l_{k(t)})}}(\boldsymbol{h}_t, \boldsymbol{x}_t)\rangle &\leq \min\left\{C, \ 4\beta_T \|\boldsymbol{\phi}_{V^{(l_{k(t)})}}(\boldsymbol{h}_t, \boldsymbol{x}_t)\|_{\boldsymbol{\Sigma}_t^{-1}}\right\}. \\
&\leq \min\left\{\beta_T, \ 4\beta_T \|\boldsymbol{\phi}_{V^{(l_{k(t)})}}(\boldsymbol{h}_t, \boldsymbol{x}_t)\|_{\boldsymbol{\Sigma}_t^{-1}}\right\}.
\end{aligned}$$

By Lemma 8, we have

$$\begin{aligned}
\sum_{t=0}^{T} &\langle \boldsymbol{\theta}^\ell - \boldsymbol{\theta}_t, \boldsymbol{\phi}_{V^{(l_{k(t)})}}(\boldsymbol{h}_t, \boldsymbol{x}_t)\rangle \\
&\leq 4\beta_T \sum_{t=0}^{T} \min\left\{1, \ \|\boldsymbol{\phi}_{V^{(l_{k(t)})}}(\boldsymbol{h}_t, \boldsymbol{x}_t)\|_{\boldsymbol{\Sigma}_t^{-1}}\right\} \\
&\leq 4\beta_T \sqrt{T \cdot \left(\sum_{t=0}^{T} \min\left\{1, \|\boldsymbol{\phi}_{V^{(l_{k(t)})}}(\boldsymbol{h}_t, \boldsymbol{x}_t)\|_{\boldsymbol{\Sigma}_t^{-1}}\right\}\right)} \\
&\leq 4\beta_T \sqrt{T \cdot \left[2d\log\left(\frac{\mathrm{tr}(\lambda \boldsymbol{I}) + TC^2 d}{d}\right) - \log\det(\lambda \boldsymbol{I})\right]} \\
&\leq 4\beta_T \sqrt{2Td \cdot \log\left(1 + \frac{TC^2}{\lambda}\right)}.
\end{aligned}$$

Next, we will bound the other part. By plugging in $1 - \nu = 1/(\lambda \cdot t'_{k(t)})$, we have

$$\sum_{t=0}^{T}\left(\frac{\nu}{\lambda \cdot t'_{k(t)}} + (1-\nu) \cdot C\right) \leq \sum_{t=0}^{T} \frac{C+1}{\lambda \cdot t'_{k(t)}} = (C+1)\sum_{t=0}^{T} \frac{1}{\lambda \cdot t'_{k(t)}}.$$

Considering the iteration triggering criteria, we get

$$t'_{k(t)+1} \leq 2t'_{k(t)} + \lambda.$$

Then, we can conclude that

$$\begin{aligned}
\sum_{t=0}^{T}\left(\frac{\nu}{\lambda \cdot t'_{k(t)}} + (1-\nu) \cdot C\right) &\leq \sum_{k=0}^{K}(C+1) \cdot \left(\frac{1}{\lambda} + \frac{1}{t'_k}\right) \\
&\leq 2(K+1) \cdot (C+1) \\
&= 2(C+1) \cdot \left(2d\log\left(1 + \frac{TC^2}{\lambda}\right) + \log\left(1 + \frac{T}{\lambda}\right) + 1\right)
\end{aligned}$$

By combining the two bounds, we get

$$\text{①} \le 4\beta_T \sqrt{2Td \cdot \log\left(1 + \frac{TC^2}{\lambda}\right)} + 2(C+1) \cdot \left(2d\log\left(1 + \frac{TC^2}{\lambda}\right) + \log\left(1 + \frac{T}{\lambda}\right) + 1\right).$$

$\square$

**Lemma 5** *Suppose that $T \ge 2$, $a \ge 1$ and $T \le k\log^2(aT)$ for all large enough $k$. Then, there exists $\eta = \eta(a)$ such that $T \le \eta \cdot k\log^2(ak)$ for all large enough $k$, i.e., $T = \mathcal{O}(k\log^2(ak))$.*

*Proof:* We prove the above lemma by contrapositive. Suppose that there doesn't exist such an $\eta$. Then, we will have, for all large enough $k$,

$$T \ge b_k \cdot k\log^2(ak),$$

where $\{b_k\}_{k=1}^{\infty}$ is a sequence with $\lim_{k\to+\infty} b_k = +\infty$. The above inequality also implies that

$$b_k \le \frac{T}{k\log^2(ak)} \le \frac{\log^2(aT)}{\log^2(ak)}.$$

Now, let's consider the following

$$\log^2(aT) \le \log^2(ak \cdot \log^2(aT)) = (\log(ak) + \log\log^2(aT))^2.$$

By the inequality $(a+b)^2 \le 2a^2 + 2b^2$, we get

$$\log^2(aT) \le 2\log^2(ak) + 2\log^2(\log^2(aT)).$$

Since $aT \ge 2$, we will have

$$\log^2(\log^2(aT)) \le \frac{1}{4}\log^2(aT) \quad \Rightarrow \quad \log^2(aT) \le 2\log^2(ak) + \frac{1}{2}\log^2(aT).$$

Therefore, we can get

$$\frac{1}{2}\log^2(aT) \le 2\log^2(ak) \quad \Rightarrow \quad b_k \le \frac{\log^2(aT)}{\log^2(ak)} \le 4,$$

which leads to a contradiction with $\lim_{k\to+\infty} b_k = +\infty$. Hence, we have $T = \mathcal{O}(k\log^2(ak))$. $\square$

## F Proof of Theorem 2

**Theorem 2** *Under Assumptions 1 and 2, if the confidence set $\mathcal{C}_t$ is constructed according to Lemma 1 with $C = \mathcal{O}(C_\star)$, $\lambda = 1/\delta_0^2$, then with probability at least $1 - 3\delta$, the total cost incurred by running Algorithm 1 for epiphany learners with $\gamma < 1$, is upper bounded by*

$$\mathcal{O}\left(C_\star \cdot \left(1 + d\sqrt{\log\left(1 + \frac{C_\star^2\delta_0^2\log\delta}{\log\gamma}\right)}\right) \cdot \sqrt{\frac{\log\delta}{\log\gamma}\log\left(\frac{C_\star\log\delta}{\delta\log\gamma}\right)}\right). \tag{8}$$

*Proof:* The proof for the epiphany learner case mostly follows from the proof of the non-epiphany learner case, i.e., Theorem 1. In the same way, we can still decompose the cost as in Theorem 1. The only differences are in the bound of ① in and the upper bound on $T$.

$$\sum_{t=0}^{T} c(\boldsymbol{h}_t, \boldsymbol{x}_t) \le \underbrace{\sum_{t=0}^{T}\left[c(\boldsymbol{h}_t, \boldsymbol{x}_t) + \mathbb{P}_{\boldsymbol{\theta}^\ell}V_{k(t)}(\boldsymbol{h}_t, \boldsymbol{x}_t) - V_{k(t)}(\boldsymbol{h}_t)\right]}_{①} + \underbrace{\sum_{t=0}^{T}\left[V_{k(t)}(\boldsymbol{h}_{t+1}) - \mathbb{P}_{\boldsymbol{\theta}^\ell}V_{k(t)}(\boldsymbol{h}_t, \boldsymbol{x}_t)\right]}_{②}$$

$$+ 2dC\log\left(1 + \frac{TC^2}{\lambda}\right) + C\log\left(1 + \frac{2T}{\lambda}\right) + C.$$

In analogy to Lemma 4, we can get the following bound for ①,

$$
① = \sum_{t=0}^{T} \left[ c(\boldsymbol{h}_t, \boldsymbol{x}_t) + \mathbb{P}_{\boldsymbol{\theta}^\ell} V_{k(t)}(\boldsymbol{h}_t) - V_{k(t)}(\boldsymbol{h}_t) \right]
$$

$$
\leq \sum_{t=0}^{T} \langle \boldsymbol{\theta}^\ell - \boldsymbol{\theta}_t, \boldsymbol{\phi}_{V^{(l_{k(t)})}}(\boldsymbol{h}_t, \boldsymbol{x}_t) \rangle + \sum_{t=0}^{T} \left( \frac{\gamma}{\lambda \cdot t'_{k(t)}} + (1-\gamma) \cdot C \right)
$$

$$
\leq 4\beta_T \sqrt{2Td \cdot \log\left(1 + \frac{TC^2}{\lambda}\right)} + \sum_{t=0}^{T} \left( \frac{\gamma}{\lambda \cdot t'_{k(t)}} + (1-\gamma) \cdot C \right).
$$

In the following, we will bound the r.h.s term in the above equation in a similar way to the proof in Lemma 4,

$$
\sum_{t=0}^{T} \left( \frac{\gamma}{\lambda \cdot t'_{k(t)}} + (1-\gamma) \cdot C \right)
$$

$$
\leq \sum_{k=0}^{K} \gamma \cdot \left( \frac{1}{\lambda} + \frac{1}{t'_k} \right) + (1-\gamma) \cdot T \cdot C
$$

$$
\leq 2\gamma \cdot \left( 2d \log\left(1 + \frac{TC^2}{\lambda}\right) + \log\left(1 + \frac{T}{\lambda}\right) \right) + (1-\gamma) \cdot T \cdot C.
$$

Then, by plugging in the above bounds, we get

$$
① \leq 4\beta_T \sqrt{2Td \cdot \log\left(1 + \frac{TC^2}{\lambda}\right)} + 2\gamma \cdot \left( 2d \log\left(1 + \frac{TC^2}{\lambda}\right) + \log\left(1 + \frac{T}{\lambda}\right) \right) + (1-\gamma) \cdot T \cdot C.
$$

The bound for ② in Theorem 1 still holds with probability at least $1 - \delta$. Hence, we can merge all the terms and simply them into

$$
\sum_{t=0}^{T} c(\boldsymbol{h}_t, \boldsymbol{x}_t) \leq 4\beta_T \sqrt{2Td \cdot \log\left(1 + \frac{TC^2}{\lambda}\right)} + 9Cd \log\left(1 + \frac{TC^2}{\lambda}\right) + 2C\sqrt{2T \log\left(\frac{T}{\delta}\right)}
$$

$$
+ (1-\gamma) \cdot T \cdot C + C
$$

$$
= \mathcal{O}\left( C \cdot \left( 1 + d\sqrt{\log\left(1 + TC^2 \delta_0^2\right)} \right) \cdot \sqrt{T \cdot \log\left(TC/\delta\right)} + (1-\gamma) \cdot T \cdot C \right)
$$

In the next, it's easy to show that, with probability at least $1 - \delta$, the following holds[4]

$$
T = \mathcal{O}(\log(\delta)/\log(\gamma)).
$$

Lastly, since $C = \mathcal{O}(C_\star)$, and plugging in the value of $T$, we have the following hold with probability at least $1 - 3\delta$ by applying the union bound over the three events (i.e., Lemma 1, bounding ② and bounding $T$),

$$
\sum_{t=0}^{T} c(\boldsymbol{h}_t, \boldsymbol{x}_t) = \mathcal{O}\left( C_\star \cdot \left( 1 + d\sqrt{\log\left(1 + \frac{C_\star^2 \delta_0^2 \log \delta}{\log \gamma}\right)} \right) \cdot \sqrt{\frac{\log \delta}{\log \gamma} \log\left(\frac{C_\star \log \delta}{\delta \log \gamma}\right)} \right).
$$

$\square$

# G   Proof of Theorem 3

**Theorem 3** *For $\epsilon$-approximate teachable epiphany learners as defined in Definition 2, if $\|\boldsymbol{\theta}_0 - \boldsymbol{\theta}^\star\|_2 \leq \delta_0$, the cost function is bounded from above by $c_{max}$, the confidence set $\mathcal{C}_t$ is constructed according to Lemma 1*

---

[4]Without the loss of generality, we assume $\log(\delta)/\log(\gamma) \geq 1$.

with $C = \mathcal{O}(\epsilon\gamma c_{max}/(1-\gamma)^2 + C_\star)$, and if $\lambda = 1/\delta_0^2$, then with probability at least $1 - 3\delta$, the teaching cost incurred by running Algorithm 1 is upper bounded by

$$\mathcal{O}\left(C \cdot \left(1 + d\sqrt{\log\left(1 + \frac{C^2\delta_0^2\log\delta}{\log\gamma}\right)}\right) \cdot \sqrt{\frac{\log\delta}{\log\gamma}\log\left(\frac{C\log\delta}{\delta\log\gamma}\right) \cdot \frac{\epsilon\log\delta}{\log\gamma}C}\right). \tag{9}$$

*Proof:* The proof for $\epsilon$-approximate teachable epiphany learner also follows from the proof of Theorem 1 and Theorem 2. However, to make the similar proof work, we have to bound the maximum value of the value function under $\mathcal{M}_{\boldsymbol{\theta}^\star}$. To show this, by Lemma 10 and Definition 2, we have

$$\|V^\star(\cdot|\boldsymbol{\theta}^\star) - V^\star(\cdot)\|_\infty \leq \frac{\gamma c_{\max}\epsilon}{(1-\gamma)^2},$$

where we use $V^\star(\cdot|\boldsymbol{\theta}^\star)$ and $V^\star(\cdot)$ to denote the optimal value function under the approximate MDP $\mathcal{M}_{\boldsymbol{\theta}^\star}$ and the true MDP $\mathcal{M}$, respectively. Therefore, we can conclude that

$$\|V^\star(\cdot|\boldsymbol{\theta}^\star)\|_\infty \leq C_\star + \frac{\gamma c_{\max}\epsilon}{(1-\gamma)^2}.$$

Together with the optimism principle in Algorithm 2, recall that

$$\eta_t = V_{k(t)} - \langle\boldsymbol{\phi}_{V_{k(t)}}(\boldsymbol{h}_t, \boldsymbol{x}_t), \boldsymbol{\theta}^\star\rangle.$$

We will have $\eta_t$ is $(C_\star + \frac{\gamma c_{\max}\epsilon}{(1-\gamma)^2})$-sub-Gaussian. Therefore, by choosing $C = C_\star + \frac{\gamma c_{\max}\epsilon}{(1-\gamma)^2}$ as assumed, we will have the following holds with probability at least $1 - \delta$ by following the same proof as in Lemma 1,

$$\boldsymbol{\theta}^\star \in \mathcal{C}_t \cap \mathcal{B}.$$

Condition on the above event, the same teaching cost decomposition in Theorem 1 still holds,

$$\sum_{t=0}^{T} c(\boldsymbol{h}_t, \boldsymbol{x}_t) \leq \underbrace{\sum_{t=0}^{T}\left[c(\boldsymbol{h}_t, \boldsymbol{x}_t) + \mathbb{P}_{\boldsymbol{\theta}^\ell}V_{k(t)}(\boldsymbol{h}_t, \boldsymbol{x}_t) - V_{k(t)}(\boldsymbol{h}_t)\right]}_{\textcircled{1}} + \underbrace{\sum_{t=0}^{T}\left[V_{k(t)}(\boldsymbol{h}_{t+1}) - \mathbb{P}_{\boldsymbol{\theta}^\ell}V_{k(t)}(\boldsymbol{h}_t, \boldsymbol{x}_t)\right]}_{\textcircled{2}}$$
$$+ 2dC\log\left(1 + \frac{TC^2}{\lambda}\right) + C\log\left(1 + \frac{2T}{\lambda}\right) + C.$$

To bound $\textcircled{1}$, the idea is similar to Lemma 4. Due to the model misspecification, there will be one additional term in the bound,

$$\sum_{t=0}^{T} c(\boldsymbol{h}_t, \boldsymbol{x}_t) + \mathbb{P}V_{k(t)}(\boldsymbol{h}_t, \boldsymbol{x}_t) - Q_{k(t)}(\boldsymbol{h}_t, \boldsymbol{x}_t)$$
$$= \sum_{t=0}^{T} c(\boldsymbol{h}_t, \boldsymbol{x}_t) + \mathbb{P}V_{k(t)}(\boldsymbol{h}_t, \boldsymbol{x}_t) - \mathbb{P}_{\boldsymbol{\theta}^\star}V_{k(t)}(\boldsymbol{h}_t, \boldsymbol{x}_t) + \mathbb{P}_{\boldsymbol{\theta}^\star}V_{k(t)}(\boldsymbol{h}_t, \boldsymbol{x}_t) - Q_{k(t)}(\boldsymbol{h}_t, \boldsymbol{x}_t)$$
$$= \sum_{t=0}^{T} \underbrace{\left(c(\boldsymbol{h}_t, \boldsymbol{x}_t) + \mathbb{P}_{\boldsymbol{\theta}^\star}V_{k(t)}(\boldsymbol{h}_t, \boldsymbol{x}_t) - Q_{k(t)}(\boldsymbol{h}_t, \boldsymbol{x}_t)\right)}_{\clubsuit} + \underbrace{[\mathbb{P} - \mathbb{P}_{\boldsymbol{\theta}^\star}]V_{k(t)}(\boldsymbol{h}_t, \boldsymbol{x}_t)}_{\heartsuit}.$$

The bound of the $\clubsuit$ term is still the same as it in Theorem 2, and the bound for the term $\heartsuit$ is

$$[\mathbb{P} - \mathbb{P}_{\boldsymbol{\theta}^\star}]V_{k(t)}(\boldsymbol{h}_t, \boldsymbol{x}_t) \leq C \cdot \epsilon.$$

By putting the two bounds together we get

$$\textcircled{1} \leq 4\beta_T\sqrt{2Td \cdot \log\left(1 + \frac{TC^2}{\lambda}\right)} + 2\gamma \cdot \left(2d\log\left(1 + \frac{TC^2}{\lambda}\right) + \log\left(1 + \frac{T}{\lambda}\right)\right)$$
$$+ (1-\gamma)\cdot T \cdot C + \epsilon \cdot T \cdot C.$$

The bound for ②  in Theorem 1 still holds here with probability at least $1 - \delta$. Hence, we can merge all the terms and simply them into

$$\sum_{t=0}^{T} c(\boldsymbol{h}_t, \boldsymbol{x}_t) \leq 4\beta_T \sqrt{2Td \cdot \log\left(1 + \frac{TC^2}{\lambda}\right)} + 9Cd \log\left(1 + \frac{TC^2}{\lambda}\right) + 2C\sqrt{2T \log\left(\frac{T}{\delta}\right)}$$
$$+ (1 - \gamma) \cdot T \cdot C + C + \epsilon \cdot T \cdot C$$

Following the proof in Theorem 2, we have the following hold with probability at least $1 - \delta$,

$$T = \mathcal{O}(\log(\delta)/\log(\gamma)).$$

By applying the union bound for the three events (i.e., Lemma 1, bounding ② and bounding $T$), and plugging in the above $T$ and $C = \mathcal{O}(C_\star + \frac{\gamma c_{\max}\epsilon}{(1-\gamma)^2})$, we can get the final bound for the teaching cost, with probability at least $1 - 3\delta$,

$$\sum_{t=0}^{T} c(\boldsymbol{h}_t, \boldsymbol{x}_t) = \mathcal{O}\left(C \cdot \left(1 + d\sqrt{\log\left(1 + \frac{C^2\delta_0^2 \log \delta}{\log \gamma}\right)}\right) \cdot \sqrt{\frac{\log \delta}{\log \gamma}} \log\left(\frac{C \log \delta}{\delta \log \gamma}\right) + \frac{\epsilon \log \delta}{\log \gamma} C\right).$$

$\square$

## H  Additional Theorems and Lemmas

**Lemma 6 (Abbasi-Yadkori et al. (2011))** *Let $\{\mathcal{F}_t\}_{t=0}^{\infty}$ be a filtration. Let $\{\eta_t\}_{t=1}^{\infty}$ be a real-valued stochastic process such that $\eta_t$ is $\mathcal{F}_t$-measurable and $\eta_t$ is conditionally B-sub-Gaussian. Let $\{\boldsymbol{\phi}_t\}_{t=1}^{\infty}$ be an $\mathbb{R}^d$-valued stochastic process such that $\boldsymbol{\phi}_t$ is $\mathcal{F}_{t-1}$-measurable. Assume that $\boldsymbol{\Sigma}$ is a $d \times d$ positive definite matrix. For any $t \geq 0$, define*

$$\boldsymbol{\Sigma}_t = \boldsymbol{\Sigma} + \sum_{i=1}^{t} \boldsymbol{\phi}_i \boldsymbol{\phi}_i^\top, \quad \boldsymbol{s}_t = \sum_{i=1}^{t} \eta_i \boldsymbol{\phi}_i.$$

*Then, for any $\delta > 0$, with probability at least $1 - \delta$, for all $t \geq 0$,*

$$\|\boldsymbol{\Sigma}_t^{-1/2} \boldsymbol{s}_t\|_2 \leq B\sqrt{2 \log\left(\frac{\det(\boldsymbol{\Sigma}_t)^{1/2}}{\delta \cdot \det(\boldsymbol{\Sigma})^{1/2}}\right)}.$$

**Lemma 7 (Abbasi-Yadkori et al. (2011))** *Suppose that $\boldsymbol{\phi}_1,...,\boldsymbol{\phi}_t \in \mathbb{R}^d$ and for any $1 \leq s \leq t$, we have $\|\boldsymbol{\phi}_s\| \leq L$. Let $\boldsymbol{\Sigma}_t = \lambda \boldsymbol{I} + \sum_{s=1}^{t} \boldsymbol{\phi}_s \boldsymbol{\phi}_s^\top$ for some $\lambda > 0$. Then,*

$$\det(\boldsymbol{\Sigma}_t) \leq (\lambda + tL^2/d)^d.$$

**Lemma 8 (Abbasi-Yadkori et al. (2011))** *Let $\{\boldsymbol{\phi}_t\}_{t=1}^{\infty}$ be in $\mathbb{R}^d$, and $\|\boldsymbol{\phi}_t\| \leq L$ for any $t$. Then, for $\boldsymbol{\Sigma}_t = \lambda \boldsymbol{I} + \sum_{s=1}^{t} \boldsymbol{\phi}_s \boldsymbol{\phi}_s^\top$, we will have*

$$\sum_{s=1}^{t} \min\left\{1, \|\boldsymbol{\phi}_s\|_{\boldsymbol{\Sigma}_{s-1}^{-1}}\right\} \leq 2\left[d \log\left(\frac{\text{tr}(\lambda \boldsymbol{I}) + tL^2}{d}\right) - \log \det(\lambda \boldsymbol{I})\right].$$

**Lemma 9 (Min et al. (2021))** *For a transition function $\mathbb{P}$, a sequence of bounded and non-negative value functions $\{V_k\}_{k=1}^{K}$ under $\mathbb{P}$, and a state action sequence $\{(\boldsymbol{h}_t, \boldsymbol{x}_t)\}_{t=1}^{T}$, where $\|V_k\|_\infty \leq C$ and $\boldsymbol{h}_{t+1} \sim \mathbb{P}[\cdot|\boldsymbol{h}_t, \boldsymbol{x}_t]$, we have the following hold with probability at least $1 - \delta$,*

$$\sum_{t=0}^{T} \left[V_{k(t)}(\boldsymbol{h}_t) - \mathbb{P}V_{k(t)}(\boldsymbol{h}_t, \boldsymbol{x}_t)\right] \leq 2C\sqrt{2T \log\left(\frac{T}{\delta}\right)}.$$

**Lemma 10 (Csáji and Monostori (2008))** *For two discounted MDPs with discounting factor $\gamma$, if they differ only in the transition functions, denoted by $\mathbb{P}_1$ and $\mathbb{P}_2$. If their corresponding optimal value functions are $V_1^\star$ and $V_2^\star$, respectively, and the cost function is bounded from above by $c_{max}$, then*

$$\|V_1^\star - V_2^\star\|_\infty \leq \frac{\gamma c_{max}}{(1-\gamma)^2} \|\mathbb{P}_1 - \mathbb{P}_2\|_\infty.$$

