# OpenReview forum: "Iterative Machine Teaching for Black-Box Markov Learners"
_TMLR — Rejected by TMLR_

### Review · Reviewer_gj6H · 2024-03-09

**Summary Of Contributions:**

The research introduces a versatile machine teaching model with parametric Markov learners, offering a substitute for various existing learner models to characterize learner transition dynamics and enabling the estimation of learner dynamics from data. Additionally, it establishes a natural connection between Markov learners and concepts in behavioral science and educational research such as epiphany and non-epiphany learning. Rigorous analyses within the proposed teaching framework highlight teaching costs for different scenarios, demonstrating polynomial growth in optimal teaching cost and feature dimension d under linear dynamics, while revealing the non-feasibility of teaching under non-linear dynamics and providing teachability conditions for linear teaching costs. Furthermore, a numerical case study illustrates the practical application of the algorithm, supplemented with guidelines for setting hyperparameters to facilitate implementation.

**Audience:**

Yes

**Broader Impact Concerns:**

There are no broader impact concerns from my perspective.

**Claims And Evidence:**

Yes

**Requested Changes:**

Please modify the content according to the above.

**Strengths And Weaknesses:**

Strengths: This paper considers several sequential learner models that have Markov property. The parametric model considered includes examples such as Preference-based version space, Gradient-based method, Skill-based method and Memory-based method. It provides the new Black-box Teaching Algorithm for both Non-Epiphany and Epiphany Learners. The theoretical results are able to bound the teaching cost for both two types of learners and the numerical study supports our theory.

Weakness:
1. One of the major contributions this paper offers is that the Parametric Markov Learners covers several learners with different formats. However, the theoretical results Theorem 1 and Theorem 2 seem to not apply for these examples listed. The significance of the theorems are diminished because of that. Please let me know if I misunderstand something.

2. The MDP setting discussed in Section 4 with target state $h^*$ seems to be closed related the goal-conditioned RL or stochastic shortest path, which has been extensively studied in the existing literature. However, this paper has no discussion about it in the related work section, a lot of literature discussion is missing.

3. The teachable Linear Markov Learners that is defined in the Assumption 1 is essentially the linear mixture model that studied previously. It should be stated clearly via the citing the appropriate literature.

4. I don't understand how to interpret Theorem 2 in terms of sample complexity. Concretely, since $C^*$ is the optimal cost, the bound translate to $V^\pi-V^\star\leq O(C^* d)$. Since $C = O(C^*)$, it automatically holds that $V^\pi-V^\star\leq O(C^*)$, therefore it is unclear what the contribution of the theorem is.

---

### Review · Reviewer_MznR · 2024-03-12

**Summary Of Contributions:**

The submission considers solving a machine-teaching problem by reformulating it as a standard MDP problem. Once converted, the problem becomes equivalent to reinforcement learning in the infinite-horizon MDP (either with or without a discount factor), hence existing solutions can be employed (e.g., Jaksch et al. 2010). The only difference from (Jaskch et al., 2010) is that the paper considers (misspecified) linear MDP with linear function approximation.

**Audience:**

Yes

**Broader Impact Concerns:**

I do not see any direct broader impact in this work.

**Claims And Evidence:**

Yes

**Requested Changes:**

Minor comments

-	Your objective cost function (2) does not involve the discount factor $\gamma$.

-	Theorem 3 – (9): shouldn’t it be the addition, not multiplication?

-	Where is the result for the undiscounted setting? ($\gamma=1$)

**Strengths And Weaknesses:**

High-Level

Since the paper is about machine teaching, I think the authors should have focused more on the machine teaching aspect. The current draft focuses too much on the technical aspects of solving the infinite-horizon linear-MDP problem, for which numerous papers have already been published in the RL literature (e.g., [1,2]). I would say the most interesting part of this paper is Section 3.1, which is not so much related to the main content of the paper since the authors assume the linear MDP structure. This is quite disappointing as a machine teaching paper.

[1] Wu et al., Nearly Minimax Optimal Regret for Learning Infinite-horizon Average-reward MDPs with Linear Function Approximation, AISTATS 2022.

[2] Zhou et al., Provably Efficient Reinforcement Learning for Discounted MDPs with Feature Mapping, ICML 2021.



Main Comments

- I do not understand the complexity results stated in the theorems. Why do you have $\log \gamma$ in the denominator? How does it compare to standard $d\sqrt{T}$ regret bound in linear MDPs?

- Why do you need $c_min > 0$? How do the results change if you do not have any costs and only care about reaching $h^*$?
What is the role of initialization, can we just assume $\theta_0=0$?

- I do not think having a shifted regularizer $\lambda \|\theta - \theta_0\|$ makes any major differences from standard analysis with linear MDP assumptions (e.g., Jin et al., 2020, Zhou et al., 2021). If so, could you elaborate on it? I would like to understand the major challenge that existing algorithms could not address.

---

### Review · Reviewer_7RcT · 2024-03-12

**Summary Of Contributions:**

The authors propose a machine teaching approach for learners with unknown learning dynamics. The unknown dynamics are modeled as an MDP, which they claim captures a wide variety of models in the literature. Assuming this is an MDP problem, the goal is to find an optimal policy, which is well known in the reinforcement learning literature. The authors propose a connection to epiphany learning, the relevance of which I do not understand. The paper concludes with numerical studies.

**Audience:**

No

**Claims And Evidence:**

No

**Requested Changes:**

The paper has a number of limitations that make it hard to understand and of unclear contribution to the literature. First, the authors are not clear about what the learning setting is. What does the learner do? What does the teacher learn? How is this related to and different from prior work? Second, it appears the the learning dynamics of the learner are restricted to finite belief states. Third, the introduction of epiphany learning is confusing on a number of levels

To address the first two limitations in greater detail. The straightforward way of modeling learning dynamics seems to be via probability distributions over hypotheses. If this is the case, one's MDP has an uncountably infinite number of states. If one wants a finite number of states, the approach that seems obvious is to assume learners transition between guesses about the true hypothesis. However, this should not be time independent! If not time independent, it should depend on time which is again (at least) countably infinite. I am a bit baffled by the claims in the paper. What is the HMM that is being learned? (If it is not time stationary?) If it is time stationary, what are the latent states and why?

Regarding epiphany learning, the phenomenon has a more common, different, name of insight learning. Curiously, nothing in the formalization of the problem seems to require this idea. What is it doing here? If it is important, a tighter connection should be drawn and a reasonable literature review conducted. Why is this a *good* model of epiphany/insight learning?

The aforementioned issues call into question the goals of the paper. Indeed, if the problem truly reduces to an MDP, is there a paper to write at all? The literature on MDPs is huge, yet unreviewed. What is new here?

Additional comments:
- Table 1 is unclear to me. What are the "representation" and "model".
- Table 1 is an incomplete summary of a very large literature. There are several papers that do not rely on exact knowledge ofd the learner's belief, either through robustness to violations of common ground, or explicit formalization of the lack of common ground. (All of which appear to be uncited.)
- "However, the practical teaching scenario with unknown learner dynamics has been under-explored so far." When is the assumption that the learner's state representation is known justified?
- In this paper, we set forward a generic teaching framework capable of capturing
unknown complex learner dynamics in real-world teaching applications" See above.
- It seems important to discuss in detail previous work that formulates the problem of teaching as a POMDP. One relevant paper is cited, but there is essentially no discussion of it.
- The points about "epiphany" learning seem to be unrelated to the core proposal of the paper.
- It would seem important to discuss the literature on curriculum learning, which has relevant work.
- "Nevertheless, these machine teaching models all assume the learner’s model
is known, and are designed in an ad-hoc way" I don't understand this claim. Deep knowledge tracing seems very very similar!
- Including the related works where they are is challenging. Specifically, given what limited knowledge I have of the proposal, the reinforcement learning connections are not interpretable.
- It is notable that most of the related works do not appear in Table 1.
- The "epiphany" learners do not seem relevant to the rest of the paper.
- The authors haven't done their due diligence: It is worth noting that the papers here on epiphany learning are a tiny subset of the papers on this topic, which traditionally goes by "insight" learning.
- The first paragraph of 3.1 is not understandable to me. It seems, given the setup, that learners could *go anywhere* after receiving teaching? Is that correct? If not, why not?
- There is a good bit more modern research on incorporating human considerations into machine teaching. For example, consider work by Mike Moser and research out of DuoLingo.

**Strengths And Weaknesses:**

The paper identifies an interesting problem: learners with unknown learning dynamics. This interesting problem most commonly coexists with the problem of unknown belief states, which is not tackled in the current paper.

---

### Decision · Action_Editor_smu9 · 2024-04-27

**Recommendation:** Reject

**Comment:**

All reviewers recommended a rejection for this paper. In the private comments, they stated that they do not find this paper making any contribution to the field of reinforcement learning or machine teaching [one reviewer], or at least, its contributions remain unclear [another reviewer]. One of the reviewers suggested that a systematic review of existing work would enable greater precision in positioning the paper. Moreover, it was mentioned that a number of papers on this topic are not cited, including some work by Michael Littman and colleagues (they did not specify the title of papers, but I believe it is easy to find).

Overall, since we have such unanimously recommendations, unfortunately I cannot accept this paper. It seems that the authors may need to go back to the blackboard and significantly revise their work.

**Audience:**

I believe that the paper could be interesting to the readers of TMLR, but since the reviewers have problems appreciating the actual contributions of this work and some of them stated No to this Audience question, it appears that the paper has not been completely successful in clearly conveying the authors' ideas to the members of this venue.

**Claims And Evidence:**

No, all reviewers stated No to the question on Claims and Evidence.

**Resubmission Of Major Revision:**

The authors may consider submitting a major revision at a later time.